# ConceptMix: A Compositional Image Generation Benchmark with Controllable Difficulty

**Xindi Wu**[*]   **Dingli Yu**[*]   **Yangsibo Huang**[*]   **Olga Russakovsky**   **Sanjeev Arora**

Princeton Language and Intelligence (PLI), Princeton University
https://princetonvisualai.github.io/conceptmix/

## Abstract

Compositionality is a critical capability in Text-to-Image (T2I) models, as it reflects their ability to understand and combine multiple concepts from text descriptions. Existing evaluations of compositional capability rely heavily on human-designed text prompts or fixed templates, limiting their diversity and complexity, and yielding low discriminative power. We propose CONCEPTMIX, a scalable, controllable, and customizable benchmark which *automatically* evaluates compositional generation ability of T2I models. This is done in two stages. First, CONCEPTMIX generates the text prompts: concretely, using categories of visual concepts (e.g., objects, colors, shapes, spatial relationships), it *randomly* samples an object and $k$-tuples of visual concepts, then uses GPT-4o to generate text prompts for image generation based on these sampled concepts. Second, CONCEPTMIX evaluates the images generated in response to these prompts: concretely, it checks how many of the $k$ concepts actually appeared in the image by generating one question per visual concept and using a strong VLM to answer them. Through administering CONCEPTMIX to a diverse set of T2I models (proprietary as well as open ones) using increasing values of $k$, we show that our CONCEPTMIX has higher discrimination power than earlier benchmarks. Specifically, CONCEPTMIX reveals that the performance of several models, especially open models, drops dramatically with increased $k$. Importantly, it also provides insight into the lack of prompt diversity in widely-used training datasets. Additionally, we conduct extensive human studies to validate the design of CONCEPTMIX and compare our automatic grading with human judgement. We hope it will guide future T2I model development.

## 1 Introduction

Text-to-Image (T2I) generation, which produces images given a text prompt describing it (see Figure 1), has made remarkable progress [35, 43, 25, 33] with the rise of diffusion models [42, 17]. However, complicated scene descriptions can still trip up these models in subtle ways that are hard to measure using traditional perceptual metrics (e.g. FID [15], IS [38], LPIPS [48]) and embedding-based approaches (e.g. CLIP [34]). This has motivated new T2I evaluations. Complicated scene descriptions often involve many *visual concepts*, i.e., fundamental visual elements such as objects, colors, and spatial relationships present in the image. *Compositional* Text-to-Image (T2I) generation refers to the ability of models to generate images that accurately combine multiple visual concepts.

**Challenges in compositional T2I evaluation.** Several existing benchmarks focus on compositionality [19, 26]. But developing a comprehensive and expandable compositional T2I benchmark remains challenging for several reasons. First, it is tricky to generate prompts that effectively compose multiple visual concepts while still maintaining coherence and realism. The difficulties arising from tricky interactions increases exponentially with the number of concepts, making it difficult to manually design diverse prompts that cover a wide range of visual concepts. As a result, existing benchmarks

---

[*]Equal Contribution.

38th Conference on Neural Information Processing Systems (NeurIPS 2024) Track on Datasets and Benchmarks.

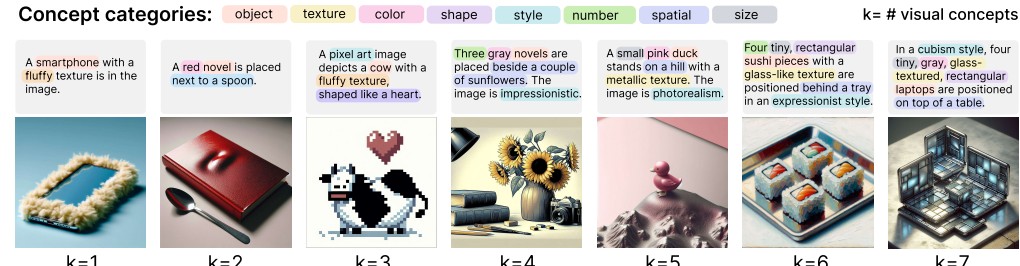

Figure 1: **Overview of CONCEPTMIX benchmark for T2I models.** Here we show some prompts generated using a different number of visual concepts. Each prompt uses a default object and a random selection of additional visual concepts from $k$ categories ($k = 1...7$, and $k = 0$ means one object, $k = 1$ means an object with one additional concept, etc.) We show images generated by DALL·E 3 [2] for these prompts. Note that the images are not part of CONCEPTMIX benchmark; the benchmark is a *distribution* of visual prompts and corresponding evaluation questions. Our CONCEPTMIX provides a scalable, controllable and customizable benchmark for compositional T2I evaluation.

Table 1: **Comparison of Compositional T2I Benchmarks.** Unlike prior benchmarks that rely on fixed templates with restricted concept categories and a constrained number of concepts per prompt, which limits the evaluation of a model's compositional generation capability, our CONCEPTMIX offers a flexible, GPT-4o-driven approach, supporting all feasible combinations of concepts and an unlimited number of concepts in each prompt.

| Benchmark | Concept Diversity | Concept Binding Method | # Concepts in Each Text Prompt |
|---|---|---|---|
| CC-500 [12] | 2 categories | Fixed template | 2 |
| ABC-6K [12] | 2 categories | Fixed template | 2 |
| Attn-Exct [6] | 4 categories | Fixed template | 2 |
| HRS-comp [1] | 2 categories | Fixed template | $\leq 3$ |
| T2I-CompBench [19] | 6 categories | Fixed template, ChatGPT augmented | $\leq 5$ |
| **CONCEPTMIX (ours)** | 8 categories | Free-form, GPT-4o generated | Unlimited |

often cover only a subset of visual concepts. Second, accurately and simultaneously evaluating multiple concepts present in the generated images is challenging. This becomes increasingly complex as the number of concepts grows, leading most evaluations to lack scalability and flexibility. They typically cap prompts to at most five concepts due to use of fixed templates for concept combination (e.g., "a {adj} {noun}"). This makes it hard to do more complex and flexible evaluations. In Tab. 1, we summarize the diversity and complexity of visual concepts and their composition in existing compositional benchmarks.

**CONCEPTMIX.** In this work, we propose CONCEPTMIX, a scalable and flexible benchmark that evaluates the compositional generation capabilities of T2I models. CONCEPTMIX operates in two key stages. First, in the prompt generation stage, CONCEPTMIX uses not fixed prompt templates but GPT-4o [31] to create prompts by combining one random object with $k$ random visual concepts. We consider eight categories of visual concepts, including objects, colors, numbers, shapes, sizes, textures, styles, and spatial relationships. The resulting prompts of CONCEPTMIX are much more diverse and complex than those of existing benchmarks (as shown in Tab. 1), which typically compose up to five visual concepts per prompt and fail to reflect the full complexity of real-world scenarios. Second, in the concept evaluation stage, CONCEPTMIX evaluates the images generated in response to these prompts by checking how many of the concepts appeared correctly in the image. This is done by generating one question per visual concept and using GPT-4o [31] to answer them. Our prompt generation pipeline also enables efficient and accurate prompt decomposition, thus we can evaluate results base on each individual concept and aggregate the results as the final score for each image. Fig. 2 provides an overview of CONCEPTMIX along with a $k = 4$ example.

Our prompt generation allows easy updating and expansion of the visual concepts to be evaluated, which is demonstrated later in §4.3 where we create variants of CONCEPTMIX. Additionally, the number of possible combinations of visual concepts grows exponentially with $k$. Thus, with a large $k$, CONCEPTMIX can generate millions of unique prompts, making it impossible for models to cheat by simply memorizing or overfitting to its training set. Thus CONCEPTMIX offers a precise and discriminative approach to identify differences in capabilities that may not be captured by traditional leaderboards or benchmarks. This provides a better understanding of a model's strengths

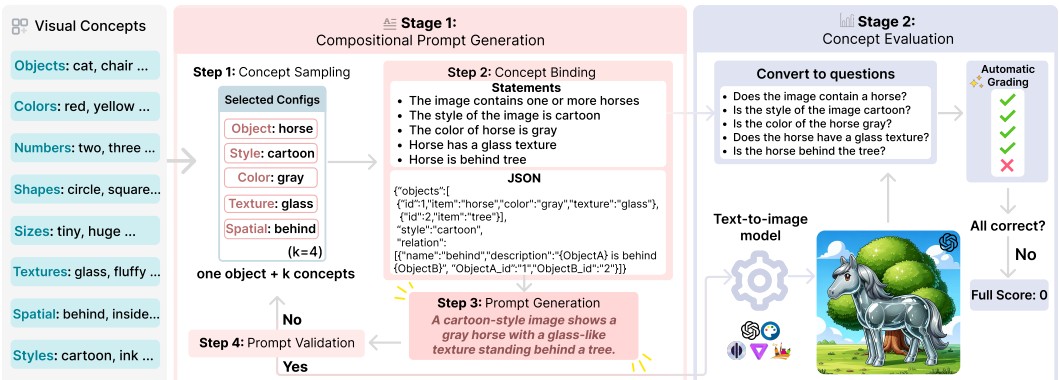

Figure 2: **CONCEPTMIX**. CONCEPTMIX consists of two main stages: 1) **Compositional Prompt Generation**: We randomly select visual concepts from 8 categories and combine them to form generation statements and intermediate JSON files with GPT-4o assistance. The statements and JSON structure are then used by GPT-4o to generate a text prompt, which, if valid, is fed into a T2I model to produce an image. 2) **Concept Evaluation**: The generated image is graded based on how well it matches with each visual concept. This is done by converting the generation statements into questions and evaluating the answers. The image receives a score of 1 if it correctly matches all concepts, and 0 if any concept is not satisfied.

and weaknesses and encourages the development of models that can combine visual concepts in meaningful and creative ways. We summarize our **main contributions** as follows:

1. We introduce CONCEPTMIX (§2), the first T2I benchmark for evaluating the compositional generation with more than five visual concepts. By dynamically combining concepts from eight different categories, CONCEPTMIX can generate a vast set of unique prompts, evaluating a model's ability to generalize beyond its training data.

2. We conduct IRB-approved human studies to validate the design of CONCEPTMIX and evaluate the effectiveness of our benchmark (§3). The study reveals high consistency between our automated grading and human evaluators. Our grading method aligns better with human preferences compared to previous approaches [26], particularly in capturing performance trends across different k values.

3. Through our systematic evaluation of eight state-of-the-art T2I models (§4), we discover: a) A consistent performance drop as $k$ increases (§4.3) with the leading proprietary model, DALL·E 3, struggling at $k = 5$. b) CONCEPTMIX clearly differentiates T2I models compared to previous compositional benchmarks [19], especially with $k \geq 2$ (§4.4). It also provides customizable evaluation by accommodating concept difficulty disparities (§4.2), resulting in easy and hard variants of CONCEPTMIX. c) Quantitative insights into models' limitations with complex prompts, with performance dropping significantly at $k = 3$ (below 25%) and at $k = 4$ (below 10%). d) The performance limitation can be traced to popular training corpora LAION [40], which we find to severely lack complex concept combinations beyond $k = 3$ (§4.5).

Our study highlights the pressing need for more challenging benchmarks to better differentiate T2I model performance and identify their limitations in compositional generation. Moreover, our findings highlight the critical need for better training data with diverse and complex visual concept combinations to improve the compositional generation capabilities of T2I models.

## 2  ConceptMix

### 2.1  Overview

CONCEPTMIX evaluates T2I models' ability to compose $k$ randomly chosen visual concepts, where $k$ controls the difficulty level. CONCEPTMIX categorizes visually interpretable concepts into eight categories, including objects, colors, numbers, and spatial relationships, etc. We define difficulty level $k$ as the number of *extra* concepts added to an image beyond a single object[1], and CONCEPTMIX($k$) is the name of the corresponding evaluation. For example, CONCEPTMIX(1) evaluates a model's ability to generate images containing a random object and another random visual concept. Since

---

[1]This approach allows us to evaluate models' capabilities beyond simple single-object generation, which is considered a well-studied problem.

Table 2: **Concept Categories in CONCEPTMIX.** We collect eight diverse visual concept categories in CONCEPTMIX to cover a wide range of visual concepts commonly used in compositional T2I generation. For each category, we provide definitions, concepts, and appearances in our text prompts.

| Category | Concepts | Example of Text Prompt containing it |
|---|---|---|
| Objects | car, chair, sushi, etc. | A **woman** is holding a **ring** in her hand |
| Colors | red, yellow, pink, etc. | A single **blue** dog is present in the image. |
| Numbers | two, three, four, etc. | The image shows exactly **four** sheep standing on a grassy field. |
| Shapes | circle, square, triangle, etc. | An oak tree with a **heart-shaped** outline stands prominently in the scene. |
| Sizes | tiny, huge, etc. | A **huge** cow is standing next to a sheep. |
| Textures | metallic, glass, fluffy, etc. | The image features a house with a **glass texture**. |
| Spatial | on top of, behind, inside, etc. | The image shows a bench with an oak tree positioned **behind** it |
| Styles | cartoon, sketch, watercolor, etc. | A **sketch** shows a single ring drawn with simple lines. |

CONCEPTMIX(0) involves no compositionality, we focus on $k \geq 1$ for the rest of the paper. By increasing $k$, we can evaluate the more challenging and realistic task of compositional generation, testing models' ability to combine multiple concepts.

We design CONCEPTMIX with two main objectives: 1) generating coherent text prompts from randomly selected concepts, and 2) automatically grading images based on complex prompts, particularly as the difficulty level ($k$) increases. The first objective is crucial for creating diverse, challenging prompts that can test T2I models' true compositional capabilities and generalization to novel concept combinations. The second objective enables us to systematically, and automatically evaluate T2I models on complex prompts. To tackle the first goal, we carefully select the sets of concepts (§2.2) and design a four-step pipeline for generating and validating the text prompts (§2.3). Building on this pipeline, we tackle the second goal by developing evaluation methods in §2.4 to grade the presence of the required concepts in the generated images and to aggregate a final evaluation score.

## 2.2 Selecting Visual Concepts

CONCEPTMIX includes eight categories of visual concepts: objects, colors, numbers, textures, shapes, sizes, styles, and spatial relationships, covering a much wider range of concepts than prior work [19] (see Tab. 2 for descriptions and examples). To ensure valid text prompts (see §2.3), we exclude concept categories where eligibility is highly object-dependent. For instance, actions are typically limited to a specific subset of objects, e.g., most objects cannot "cut", "dance" or "fly". This exclusion is crucial because our random selection of concepts, despite a filtering mechanism (see §2.3), would be less efficient if categories like actions were included.

For each category, we identify representative concepts from existing literature [19, 26] and supplement them with a diverse set generated by GPT-4. We then filter concepts that: 1) rarely combine with others (e.g., "spongy" texture), 2) are challenging for current T2I models even individually [44] (e.g., the number "6"), and 3) are difficult to judge objectively (e.g., "median" size, "minimalism" style).

## 2.3 Compositional Prompt Generation

CONCEPTMIX($k$) evaluates compositional capability by randomly sampling $k$ concepts with one object, and prompting T2I models to generate images containing all of them. This process involves four steps: 1) randomly select $k$ concept categories and choose concepts from them (**concept sampling**), 2) generate a description for each concept and create a JSON representation of the binding structure (**concept binding**), 3) generate a text prompt based on the binding structure (**prompt generation**), and 4) validate the generated text prompt using GPT-4o (**prompt validation**). Details of each step and the GPT-4o query templates are provided in Appendix C.

**Step 1: Concept Sampling.** We first sample $k + 1$ concept categories, then sample specific concepts from those categories. We always ensure that the first concept comes from the object category. The remaining $k$ concepts have a $1/4$ chance of being objects and a $3/4$ chance of being sampled from the other seven categories. This distribution helps to avoid two undesirable scenarios: (1) having most prompts contain too many objects, and (2) having most prompts contain only one object. This ensures diverse representations of concepts while maintaining a strong focus on objects, which are central to the image. We resample if there is more than one concept sampled from the style category or if the number of concepts from any category (except for the spatial category) exceeds the number of objects. This is to maintain a balanced composition and prevent any single concept category from dominating the generated image.

**Step 2: Concept Binding.** For concepts from the color, number, shape, size, or texture categories, we randomly select an object and bind the concept to it. If spatial is selected as one of the $k$ categories, we ask GPT-4o to bind each spatial concept with two objects[2]. In some cases, a concept may need a reference object to be accurately illustrated. For example, one cannot judge if an object is tiny or not if it is the only object in the image. In such cases, we also request GPT-4o to add appropriate reference objects. We formalize the binding as $k + 1$ statements (one for each concept) and a JSON object. In Fig. 2, we provide an example ($k = 4$) demonstrating the concept binding process.

**Step 3: Prompt Generation.** Given the $k + 1$ statements and the binding structure represented in JSON format, GPT-4o is asked to make up a human-annotated description of a hypothetical image that matches the statements and the JSON object. GPT-4o is instructed to avoid introducing unnecessary objects or descriptions, as detailed in the prompting template in Appendix C.

**Step 4: Prompt Validation.** Before we feed the text prompts to T2I models, we have a prompt rejection mechanism (as detailed in Appendix C.2) to validate the text prompts with GPT-4o to rule out text prompts with hard conflict between visual concepts. Note that we do not simply remove unrealistic prompts (e.g., a horse with glass texture, as shown in Fig. 2), as they can be utilized to test the creativity of T2I models. As another example, it rejects text prompts requesting a triangle-shaped person but keeps text prompts requesting a square-shaped cloud, since clouds can naturally have various abstract shapes, while a triangle-shaped person conflicts with the perceptual constraints on human form. GPT-4o is asked to provide an explanation if it considers the text prompt invalid.

## 2.4 Concept Evaluation

We evaluate the generated images from T2I models by utilizing the visual question-answering capability of GPT-4o. Specifically, for each statement used in text prompt generation, we first ask GPT-4o to generate the corresponding yes or no question based on both the statement and the text prompt, and then send the question with the generated image to GPT-4o in a new conversation and record its answer ("Yes" or "No"). We award one point for each correctly illustrated statement, so the maximum possible points is $k + 1$.

Note naively asking GPT-4o or other vision language models (VLMs) whether the generated image matches the text prompt *does not work well* from our preliminary experiments, especially when $k$ is large and the text prompts are complicated. Decomposing the text prompt is often used as an alternative for evaluating images generated from text prompts [8, 18]. However, previous decomposing methods may generate nonsensical questions when handling complex prompts [26], and thus harm their accuracy. Since the text prompts used in CONCEPTMIX are generated from given concepts, we have effectively decomposed the text prompt correctly. Although there might be additional information injected during our text prompt generation pipeline, we ensure the information injection is minimal and natural at each step. Our approach provides a reliable and precise method for evaluating the generated images based on the decomposed concepts from the original text prompt.

# 3 Human Evaluation

To evaluate the performance of our automatic grading with GPT-4o, we conducted human evaluations with 10 participants, including both experts and non-experts. The evaluation covered 14 sets: 8 models at $k = 3$ and DALL·E 3 at $k = 1$ through 7. Each set contained 25 text prompts, generated images, and questions. Each of the 350 pairs was evaluated by 5 participants. Our procedure was reviewed and approved by our internal Institutional Review Board (IRB) and we obtained participant consent. The human evaluation involves a two-step process: 1) **Image-Prompt Alignment**: participants evaluate the overall alignment between the generated image and the text prompt; 2) **Individual Questions**: they answer individual yes/no questions based on the image. Detailed evaluation instructions and qualitative analysis are in Appendix A.

**Human annotations are inconsistent and often miss details.** Our analysis reveals notable inconsistencies in human annotations. We found that in 4.52% of cases, evaluators incorrectly answered "Yes" to step 1 image-prompt alignment when their step 2 individual concept evaluations collectively indicated "No," and vice versa in 4.93% of cases. These discrepancies result in a 9% divergence rate between steps 1 and 2, showing the importance of breaking down alignment evaluation into

---

[2]If there aren't enough existing objects for binding the spatial concepts, we request GPT-4o to add objects that naturally fit into the scene.

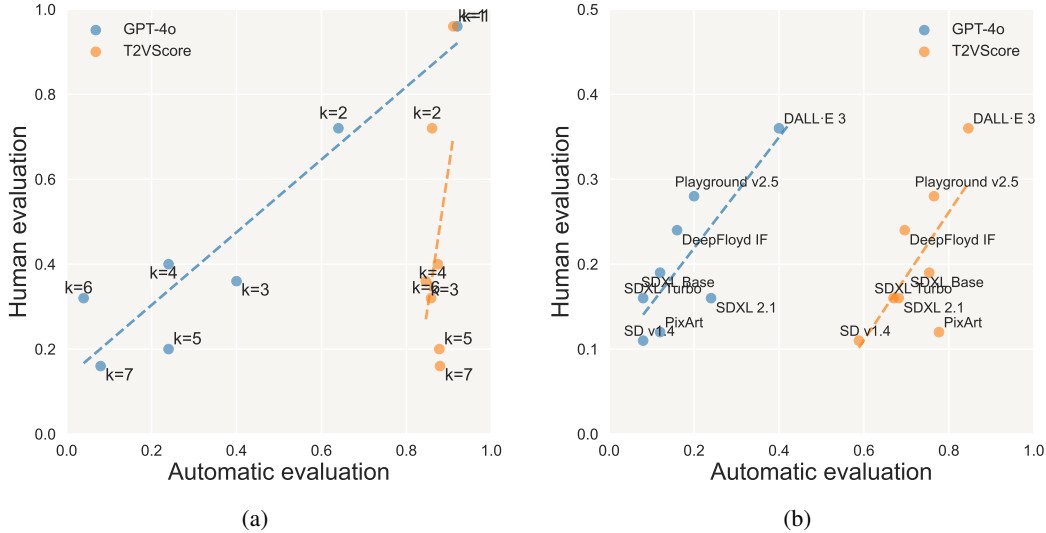

(a)  (b)

Figure 3: **Human vs. Automatic Score Correlation (Ours vs. T2VScore [26])**. (a) Correlation at various $k$ values with the DALL·E 3 model, where GPT-4o (ours) achieves a high correlation with human judgments ($r = 0.93$) compared to T2VScore ($r = 0.47$). (b) Correlation across different models at $k = 3$, with GPT-4o (ours) maintaining stronger alignment with human scores ($r = 0.81$) than T2VScore ($r = 0.69$). $r$ represents Pearson's correlation coefficient.

Table 3: **Human Evaluation on Specific Concept Category.** We show the average consistency (%) between human majority vote and GPT-4o grading across concept categories. Higher consistency percentages indicate stronger agreement with human evaluations.

| Category | Object | Color | Number | Shape | Size | Texture | Style | Spatial |
|---|---|---|---|---|---|---|---|---|
| Average Consistency (%) | 90.86 | 86.21 | 82.78 | 79.61 | 76.92 | 76.03 | 74.22 | 73.33 |

individual concept evaluation and the challenges in human evaluation, such as overlooking details or misinterpretation. We show this variability in agreement rates across different evaluation steps for DALL·E 3 in Fig. 10 in appendix A. In Fig. 3, we compare the full mark scores by GPT-4o and human scores over different settings. Human scores are the average of the human majority votes across 25 pairs. From Fig. 3a, we observe that GPT-4o is close to human scores, except for $k = 6$, the human evaluators give much higher scores than the GPT-4o. It may be caused by human oversight when the complexity of text prompts increases. Despite this, the overall trend of human scores shows a decline as $k$ increases, matching the trend of GPT-4o scores. In Fig. 3b, we observe that the human ranking is also similar to GPT-4o ranking.

**GPT-4o grader in general shows high consistency with human annotators.** We compute the consistency scores among human annotators and between human annotators and GPT-4o in Fig. 11 in Appendix A. Consistency score is defined as the ratio of two scorers giving the same score for a (prompt, image) pair among all of the (prompt, image) pairs. The average consistency score between human annotators for this task is 0.85, showing the relative subjectivity and challenge of the evaluation. The consistency score between the human majority vote and GPT-4o is 0.81, which is comparable to the inter-annotator consistency score.

**Compare with prior grading approach.** We further conduct experiments with previous state-of-the-art grading approach [26] and compare them with human preferences. As shown in Fig. 3, our grading method aligns better with human preferences. For example, in Fig. 3a, as $k$ grows, both our grading results and human majority vote results generally decrease. This trend, however, is not observed with T2VScore. Our method achieves a high correlation with human evaluations, with $r = 0.93$ for GPT-4o compared to T2VScore's $r = 0.47$. Additionally, in Fig. 3b, we observe that T2VScores remain similar across many models and do not correlate as well with human scores, with a correlation of $r = 0.69$ compared to our method's $r = 0.81$. Our method stands out by accounting for different difficulty levels and this shows that our grading approach can differentiate between various generation models and better reflect human preferences.

**Consistency Analysis Across Different Concept Categories.** We show the average consistency score of the human majority vote and GPT-4o grading results across different concept categories in

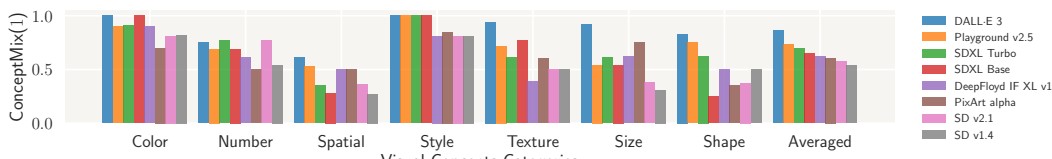

Figure 4: **Performance Across Concept Categories.** We evaluate the performance of T2I models across different concept categories. Color and style are easier, with all models achieving high scores. Performance is lower for generating specific numbers of objects and spatial relationships, with varying results for texture and size. Overall, DALL·E 3 outperforms others in all categories.

Tab. 3. These results show that GPT-4o performs relatively well across different categories, with the highest consistency observed in the object (90.86%) and color (86.21%) categories. However, as expected, the consistency is lower in categories such as spatial and style, which involve more complex spatial reasoning and style recognition tasks which are also challenging to human participants. Other categories like shape (79.61%), size (76.92%), and texture (76.03%) fall between these extremes.

## 4 Experiments

In this section, we present a systematic evaluation of eight T2I models on CONCEPTMIX, with the experimental setup detailed in §4.1. We begin by analyzing the performance of individual concept categories ($k = 1$, see §4.2) to assess how well models handle specific concept categories in isolation. Next, we evaluate the models' performance when combining multiple concept categories ($k > 1$, see §4.3), and compare CONCEPTMIX with other existing evaluation pipelines (§4.4). Finally, we explore whether common training datasets are sufficient for effective compositional generation (§4.5).

### 4.1 Experimental Setup

**Evaluated models.** We evaluate eight state-of-the-art T2I models: SD v1.4 [35], DeepFloyd IF XL v1, SD v2.1, SDXL Base [33], SDXL Turbo [39], Playground v2.5 [25], PixArt alpha [7] and DALL·E 3 [2]. We provide the details of generation configuration and compute details for our evaluation in Appendix D.

**Prompt Generation Details.** We randomly generate text prompts from CONCEPTMIX, as detailed in §2.3, and request models for generations. Each prompt includes at least one object along with $k$ additional visual concept categories. Unless specified otherwise, we randomly assign concepts from each category. We evaluate with $k \in \{1, 2, 3, 4, 5, 6, 7\}$, and for each $k$, we generate 300 text prompts to capture the variability and performance across different models.

**Concept Evaluation Details.** Given a fixed $k$, we use GPT-4o, as described in §2.4, to grade each image and determine the number of points awarded out of $k + 1$, with each point representing a required concept. We consider two grading metrics: 1) **Full-mark score**, which measures the proportion of generated images where the image correctly satisfies *all* $k + 1$ required concepts, and 2) **Concept fraction score**, which measures the average proportion of visual concepts satisfied by the generated images. Unless otherwise specified, the term 'performance' refers to full-mark score. For each model and each $k$, we report the full-mark score (Tab. 4) and concept fraction score (Appendix D.5), aggregated over 300 sampled prompts, and provide the 95% confidence interval for each score.

### 4.2 Performance on Individual Concept Categories ($k = 1$)

We begin by analyzing the performance of the models on the case $k = 1$ with each concept category, i.e., the ability to generate images of a random object and a concept within the selected category. This is the simplest form of compositional image generation. Our findings are listed as follows.

**Color and style are easiest while spatial, size, and shape are challenging.** Fig. 4 shows each model's performance across categories. A notable trend is that color and style are easier categories than others. For instance, DALL·E 3 excels in color and style, achieving perfect scores, and performs well in texture as well. However, it scores considerably lower in number and spatial categories, achieving only 0.75 and 0.61, respectively. Such findings highlight the limitations of using pixel-level similarity scores for evaluation. While these scores effectively capture style and color accuracy, they struggle to accurately reflect spatial, shape, and size. Consequently, models that perform well on these scores might still fall short in accurately generating spatial, shape, and size information.

Table 4: **Performance of Eight T2I Models on CONCEPTMIX.** We vary difficulty levels $k$ from 1 to 7 and report the full mark scores, which represent the proportion of generated images that correctly satisfy all $k + 1$ required visual concepts. As $k$ increases, all models' performance decreases, but at varying rates.

| | $k = 1$ | $k = 2$ | $k = 3$ | $k = 4$ | $k = 5$ | $k = 6$ | $k = 7$ |
|---|---|---|---|---|---|---|---|
| SD v1.4 [35] | $0.52_{\pm 0.06}$ | $0.23_{\pm 0.05}$ | $0.08_{\pm 0.04}$ | $0.03_{\pm 0.03}$ | $0.01_{\pm 0.02}$ | $0.00_{\pm 0.01}$ | $0.00_{\pm 0.01}$ |
| SD v2.1 [33] | $0.52_{\pm 0.06}$ | $0.29_{\pm 0.05}$ | $0.14_{\pm 0.04}$ | $0.06_{\pm 0.03}$ | $0.03_{\pm 0.03}$ | $0.01_{\pm 0.02}$ | $0.00_{\pm 0.01}$ |
| SDXL Turbo [39] | $0.64_{\pm 0.06}$ | $0.35_{\pm 0.06}$ | $0.18_{\pm 0.05}$ | $0.09_{\pm 0.04}$ | $0.03_{\pm 0.03}$ | $0.02_{\pm 0.02}$ | $0.01_{\pm 0.02}$ |
| PixArt alpha [7] | $0.66_{\pm 0.06}$ | $0.37_{\pm 0.06}$ | $0.17_{\pm 0.05}$ | $0.09_{\pm 0.04}$ | $0.05_{\pm 0.03}$ | $0.01_{\pm 0.02}$ | $0.01_{\pm 0.02}$ |
| DeepFloyd IF XL v1 [43] | $0.68_{\pm 0.06}$ | $0.38_{\pm 0.06}$ | $0.21_{\pm 0.05}$ | $0.09_{\pm 0.04}$ | $0.05_{\pm 0.03}$ | $0.02_{\pm 0.02}$ | $0.01_{\pm 0.02}$ |
| SDXL Base [33] | $0.69_{\pm 0.06}$ | $0.43_{\pm 0.06}$ | $0.18_{\pm 0.05}$ | $0.09_{\pm 0.04}$ | $0.05_{\pm 0.03}$ | $0.01_{\pm 0.02}$ | $0.00_{\pm 0.01}$ |
| Playground v2.5 [25] | $0.70_{\pm 0.06}$ | $0.46_{\pm 0.06}$ | $0.22_{\pm 0.05}$ | $0.10_{\pm 0.04}$ | $0.07_{\pm 0.04}$ | $0.02_{\pm 0.02}$ | $0.00_{\pm 0.01}$ |
| DALL·E 3 [2] | $0.83_{\pm 0.05}$ | $0.61_{\pm 0.06}$ | $0.50_{\pm 0.06}$ | $0.27_{\pm 0.05}$ | $0.17_{\pm 0.05}$ | $0.11_{\pm 0.04}$ | $0.08_{\pm 0.04}$ |

**Varying performance of concepts within the same category.** Fig. 5 shows the performance of Playground v2.5 across different concepts within the easiest (color) and most challenging (spatial) categories identified earlier. The performance on different concepts varies significantly. In the color category, 'red' and 'green' score higher than 'brown' and 'black'. Similarly, for spatial concepts, 'in front of' and 'right' outperform 'left' and 'bottom'. Similar variations are observed in other categories with other models, suggesting the existence of disparities in generation performance even within the same visual concept category.

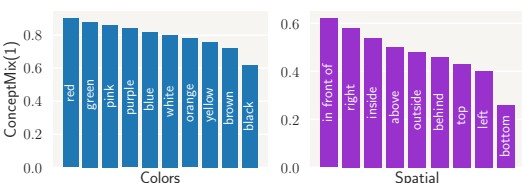

Figure 5: **Individual Concept Performance.** CONCEPTMIX scores for Playground v2.5 with $k = 1$ for colors (left) and spatial (right) concepts show performance varies within each category. More details on other categories are in Appendix D.5.

Based on the observation, we split each concept category into easy and hard subsets. We then create two variants of CONCEPTMIX: one using the easy concepts and the other using hard concepts, see Appendix C for more details.

## 4.3 Performance of Compositional Generation ($k > 1$)

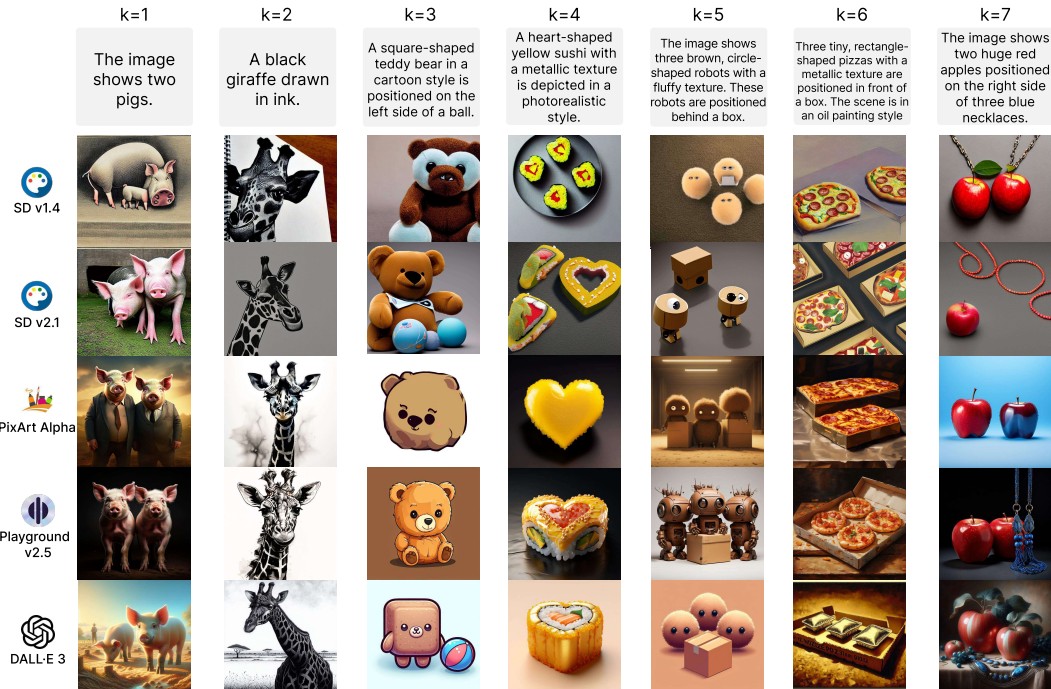

Figure 6: **Qualitative performance** of different T2I models (SD v1.4, SD v2.1, PixArt alpha, Playground v2.5, DALL·E 3) across varying levels of compositional complexity ($k = 1...7$). As prompts become more complex, image quality degrade. DALL·E 3 performs best, while SD v1.4 performs worst.

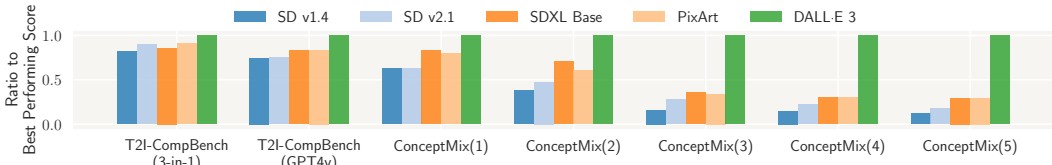

Figure 8: **CONCEPTMIX Shows Stronger Discriminative Power.** We compare five models using 3-in-1 and GPT4v scores (global prompt-level) from T2I-CompBench [19], and CONCEPTMIX with varying difficulty levels ($k$). Unlike T2I-CompBench, which shows similar scores across models, CONCEPTMIX effectively differentiates model performance, with gaps widening as $k$ increases.

**Models performance degrades when $k$ increases.** Now we examine model performance when combining multiple concept categories ($k > 1$) on our CONCEPTMIX benchmark. As shown in Tab. 4, DALL·E 3 consistently outperforms other models across all $k$ difficulty levels and can handle complex compositional tasks more effectively. As $k$ increases, all models show a significant drop in performance. Among all, the performance of SD v1.4 decreases the fastest as $k$ increases, as we can see its performance approaching zero when $k = 3$. Other models also experience performance drops but at different rates. The models can be roughly ranked by their position in the table, with DALL·E 3 being the best, and SD v1.4 being the worst. SDXL Turbo, PixArt alpha, SDXL Base, DeepFloyd IF XL v1, and Playground v2.5 have relatively close performance, with SDXL Base performing better at $k = 2$, DeepFloyd IF XL v1 and Playground v2.5 performing better at $k = 3$. We provide qualitative examples in Fig. 6 and we report the concept fraction score in Appendix D.5.

**Easy and hard variants of CONCEPTMIX.** We create two variants of CONCEPTMIX based on §4.2: one only uses the easy subsets of all categories, and the other uses the hard subsets. In Fig. 7, we plot the performance of three models on the two variants, as well as the standard CONCEPTMIX. With both variants, we again observe the degradation of model performance when $k$ increases. Furthermore, the model ranking remains consistent, indicating the robustness of CONCEPTMIX. Although the easy and hard subsets are selected based on Playground v2.5 performance on these concepts with $k = 1$, models always achieve higher scores on the easy variant compared to the hard variant.

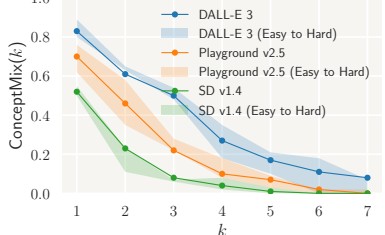

Figure 7: CONCEPTMIX($k$) drops significantly as $k$ increases, with DALL·E 3 consistently outperforming others. Shaded areas indicate the score range from easier to harder visual concepts for each $k$.

## 4.4 CONCEPTMIX has stronger discriminative power than other evaluation pipelines

We compare CONCEPTMIX with the prior compositional generation benchmark, T2I-CompBench [19], which uses a fixed template to combine at most five visual concept categories within a single prompt (see Tab. 1). While T2I-CompBench incorporates several evaluation metrics, its limited concept and prompt diversity often lead to closely clustered scores for different models, making it challenging to differentiate their performance (see Fig. 8). This lack of differentiation also hinders the identification of model limitations.

In contrast, CONCEPTMIX includes a wider range of concept categories with a total of 96 unique visual concepts and prompting variations (see Appendix C), and offers a more **precise** and **discriminative** grading approach (see Fig. 8), especially as $k$ increases. When considering $k = 1, ..., 7$ elements, the total number of combinations reaches approximately 145 billion. This offers extensive evaluation for the compositional generation of T2I models.

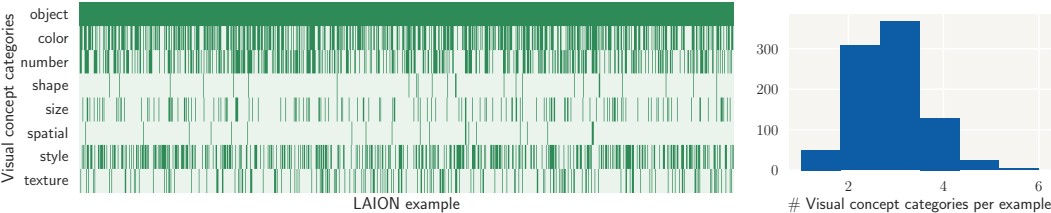

Figure 9: **Concept Diversity in LAION-5B Dataset.** Left: Heatmap of sampled captions shows colors and styles are most frequent; shapes and spatial relationships are least. Right: Most examples include 2-3 concepts.

## 4.5 Tracing the poor performance of models back to lack of diversity in training data

To further investigate the relatively poor compositional capabilities of the models, we explore whether the complexity of visual concepts in the training data might be a contributing factor. We randomly sample 1000 image captions from LAION [40], a widely used dataset for training T2I models, following ethical use guidelines for research purposes. For each caption, we use GPT-4o to identify the presence of eight visual concept categories (object, color, number, shape, size, spatial, style, and texture), with the instructions for GPT-4o provided in Appendix D. We filter out captions that did not contain objects (leaving 882 out of 1000) and plot the frequency of each concept in Fig. 9.

**Disparate concept representation in LAION-5B.** Our analysis reveals a significant disparity in the presence of different visual concepts within the LAION-5B dataset. While most captions included color (476) and style (269), only a small number contained shape (24) and spatial (20) concepts. This uneven distribution aligns with the individual visual concept performance observed in Section 4.2, suggesting that a model's proficiency in a particular visual concept might be directly influenced by the frequency of its representation in the training data.

**Limited exposure to complex concept combinations in LAION-5B.** Furthermore, we find that each example from the sampled LAION-5B collection, on average, contains only $2.75 \pm 0.90$ concept categories, with a maximum of six concepts per example. This limited exposure to complex combinations of visual concepts in the training data likely contributes to the observed difficulty models face when dealing with $k \geq 3$ (see Tab. 4).

## 5 Discussion

**Limitations.** One potential limitation of CONCEPTMIX is the potential misalignment between autograding and human grading. While our method aligns with human preference better than previous metrics, it may overlook nuances human graders capture, particularly in cases where the generated images are ambiguous. Therefore, while our grading engine offers consistent and scalable evaluation, outperforming previous approaches, it still cannot fully replicate human judgment.

**Negative Impacts.** T2I models trained on web-scale data carry inherent risks, such as privacy and copyright violations, and social bias perpetuation. Although our work focuses on the *evaluation* of the generative models, with the goal of reducing errors in generation, the downside is that CONCEPTMIX may also provide further legitimacy to generative models despite their ethical concerns.

## 6 Conclusion

Compositional capabilities are critical for T2I generation. We gave evidence that existing evaluations of compositionality, which generate prompts automatically with fixed templates, actually result in prompts with low diversity and discriminative power. We propose CONCEPTMIX, a scalable and customizable benchmark for evaluating the compositional capabilities of T2I models, including prompts from 8 visual concept categories. Our approach uses a powerful LLM in two ways to address the limitations of existing benchmarks. The first is in generating suitable prompts given a random set of visual concepts. The second is to enable automated grading of the generated image by providing a list of questions that can be used with a VLM (GPT-4o in our case) to check the correctness of the generated images. CONCEPTMIX allows generating a wide variety of prompts — the total number of possible prompts is larger than the size of popular training datasets. We find that CONCEPTMIX effectively differentiates between models, offering a more granular understanding of the strengths and weaknesses of generation models compared to traditional benchmarks.

## Acknowledgement

This material is based upon work supported by the National Science Foundation under Grant No. 2107048. Any opinions, findings, and conclusions, or recommendations expressed in this material are those of the author(s) and do not necessarily reflect the views of the National Science Foundation. DY and SA are supported by NSF and ONR. YH is supported by the Wallace Memorial Fellowship. We thank many people for their helpful discussion, feedback, and human studies listed in alphabetical order by last name: Allison Chen, Jihoon Chung, Victor Chu, Derek Geng, Luxi He, Erich Liang, Kaiqu Liang, Michel Liao, Yuhan Liu, Abhishek Panigrahi, Simon Park, Ofir Press, Zeyu Wang, Boyi Wei, David Yan, William Yang, Zoe Zager, Cindy Zhang, and Tyler Zhu from Princeton University, Zhiqiu Lin, Tiffany Ling from Carnegie Mellon University, Chiyuan Zhang from Google Research.

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

# Appendices

# A  Human Evaluation

## A.1  Human Evaluation Instructions

Here are the human evaluation instructions for participants:

---

### Human Evaluation Instructions

Your task is to evaluate the alignment between the image and the text description. Follow the steps outlined below:

**Step 1: Image-Prompt Alignment.** First, determine whether the image aligns with the description provided in the prompt. If the image aligns with the description, your answer should be 1 (yes). If the image does not align with the description, your answer should be 0 (no).

**Step 2: Individual Questions.** For each specific question listed, determine if the answer is 1 (yes) or 0 (no) based on the image. Note that once you start step 2, you can not return to step 1 to change your answers. **Example**:

**Step 1**: Image-Prompt Alignment.

**Prompt**: A photorealistic image shows a rectangle-shaped smartphone positioned in front of a table, closer to the observer. The smartphone is clearly distinguishable from the table behind it.

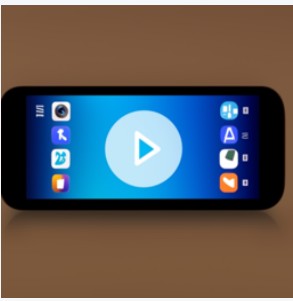

**Step 2**: Individual Questions.
**Question #1**: Does the image contain a smartphone?
**Question #2**: Is the style of the image photorealism?
**Question #3**: Is the smartphone rectangle-shaped?
**Question #4**: Is the smartphone positioned in front of the table, closer to the observer?

---

In addition to the instructions and example above, we also offer general guidance for visual concepts that may be subjective in judgment. Specifically,

**Size**  For "tiny" and "huge", judge whether the object is tiny or huge compared to its normal size in reality, which can be inferred based on the size of other objects (assuming the other objects have normal sizes).

**Style**  We define all the art styles in the rubric and provide reference images.

## A.2  Human Agreement Analysis

Fig. 10 shows the variability in agreement rates across different evaluation steps for DALL·E 3, with $k = 1$ showing high agreement between evaluation steps 1 and 2, while $k = 6$ shows lower agreement. Factors contributing to these inconsistencies may include cognitive biases, fatigue, and the complexity of the subject matter. Our experiments show the importance of a structured and granular approach in evaluation processes to improve alignment and reliability in evaluation, particularly for complex compositional capability.

## A.3  Pairwise Consistency Analysis

The average consistency score among human evaluators for this task is 0.85, showing the subjectivity and difficulty nature of the T2I evaluation. The consistency score between the human majority vote and GPT-4o is 0.81, which is similar to the inter-annotator consistency. Notably, at $k = 5$ and $k = 6$, there is a noticeable decrease in agreement among human evaluators and between humans and GPT-4o, indicating greater variability in these settings. The average human-to-human consistency score across all these models is 0.87 for $k = 3$. These findings reveal variations in human evaluations, which differ notably from automated approaches.

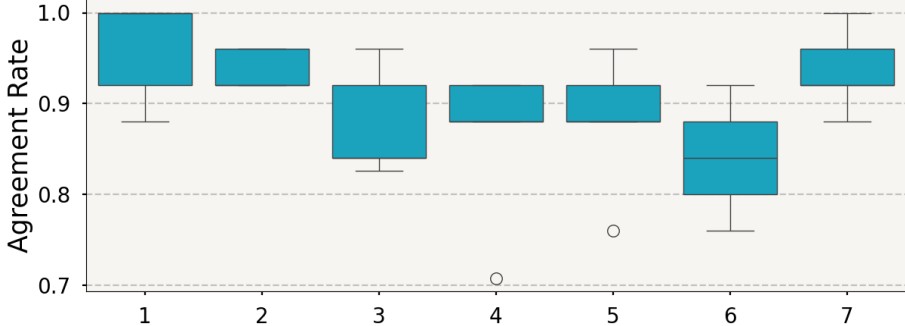

Figure 10: **Human Evaluation Agreement Rates Distribution between Step 1 and 2**. This boxplot shows the variability in evaluator agreement between evaluation steps 1 and 2 across different $k$ for DALL·E 3. Notable differences are observed between steps, with $k$=1 showing high agreement and consistency, while $k$=6 displays lower agreement and increased variability.

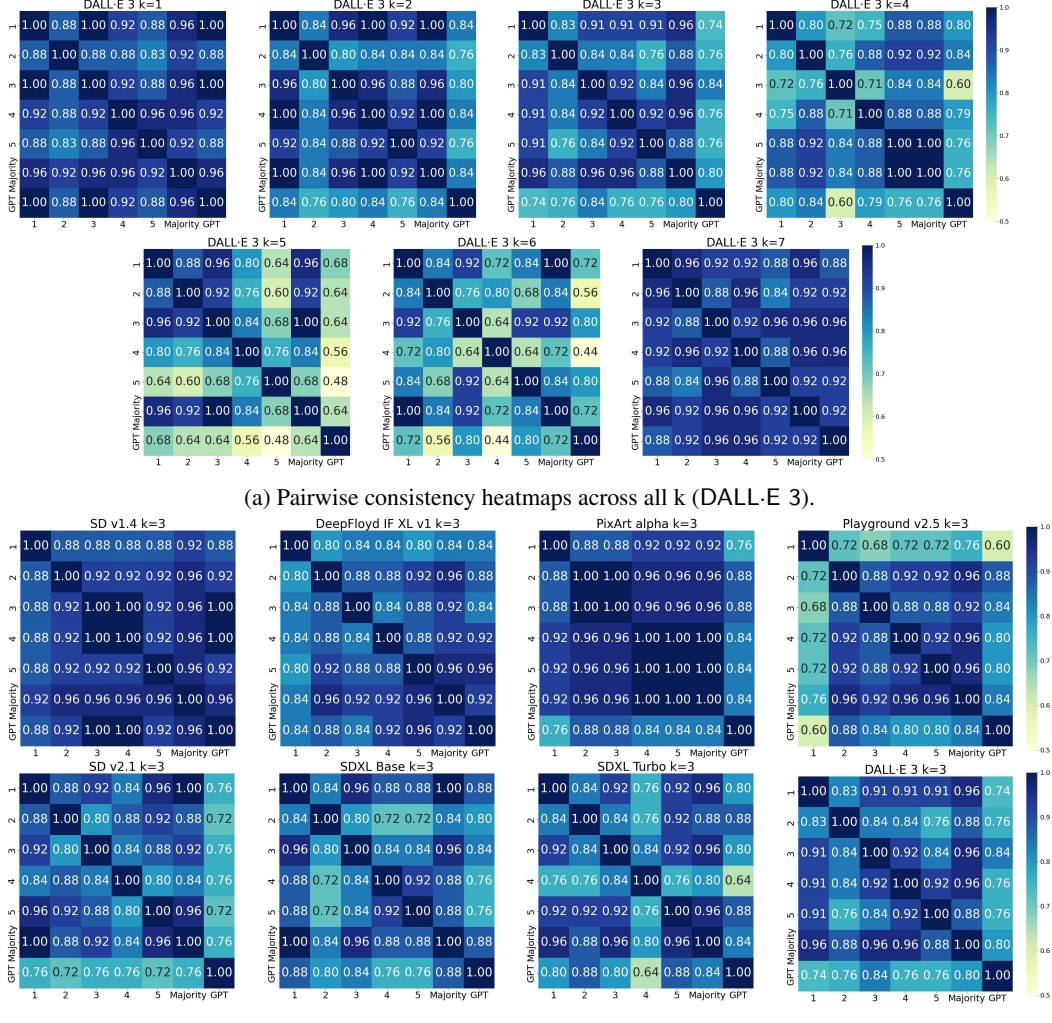

Figure 11: **Pairwise Consistency Heatmaps.** These heatmaps show the consistency between different human evaluators (**1** to **5**), human majority vote (**Majority**), and GPT-4o grading (**GPT**). The darker shades indicate higher agreement. We show the consistency heatmaps (a) across all k values for DALL·E 3 and (b) across various models. This shows that human evaluations also vary a lot with each other.

Table 5: **Human Evaluation Across** $k$ **Values.** We show the average consistency between GPT-4o and human evaluations for different $k$ values in DALL·E 3 image generation. Higher consistency percentages indicate stronger agreement with human evaluations.

| k | 1 | 2 | 3 | 4 | 5 | 6 | 7 |
|---|---|---|---|---|---|---|---|
| Average Consistency | 0.96 | 0.84 | 0.80 | 0.76 | 0.64 | 0.72 | 0.92 |

## A.4 Consistency Analysis Across k Values

To address concerns about GPT-4o's consistency in evaluating images with varying levels of complexity, we conducted a detailed analysis of its performance across different $k$ values for DALL·E 3. The results of this analysis are presented in Tab. 5. The consistency generally decreases as $k$ increases, with a dip observed at $k = 5$ (64%), reflecting the increasing complexity of the tasks. Interestingly, there is a noticeable rebound in consistency at $k = 6$ (72%) and $k = 7$ (92%), which might be attributed to the increasing complexity of compositional generation as $k$ grows. As the task becomes more challenging, the probability of generating fully correct images approaches zero. Consequently, both GPT-4o and human evaluators might converge on similar evaluation. Overall, these findings suggest that GPT-4o is capable of maintaining strong performance even as the complexity of the task varies, although some variability is observed in mid-range $k$ values.

## A.5 Qualitative Analysis

During the evaluation, we noticed several instances where human evaluators disagreed among themselves or with the GPT-4o grading method. In some cases, GPT-4o tends to be stricter in its grading. For instance, an image slightly deviating from the prompt's specifics might receive a lower score from GPT-4o, while human evaluators might overlook minor discrepancies and incorrectly grade it higher. Here we show some examples:

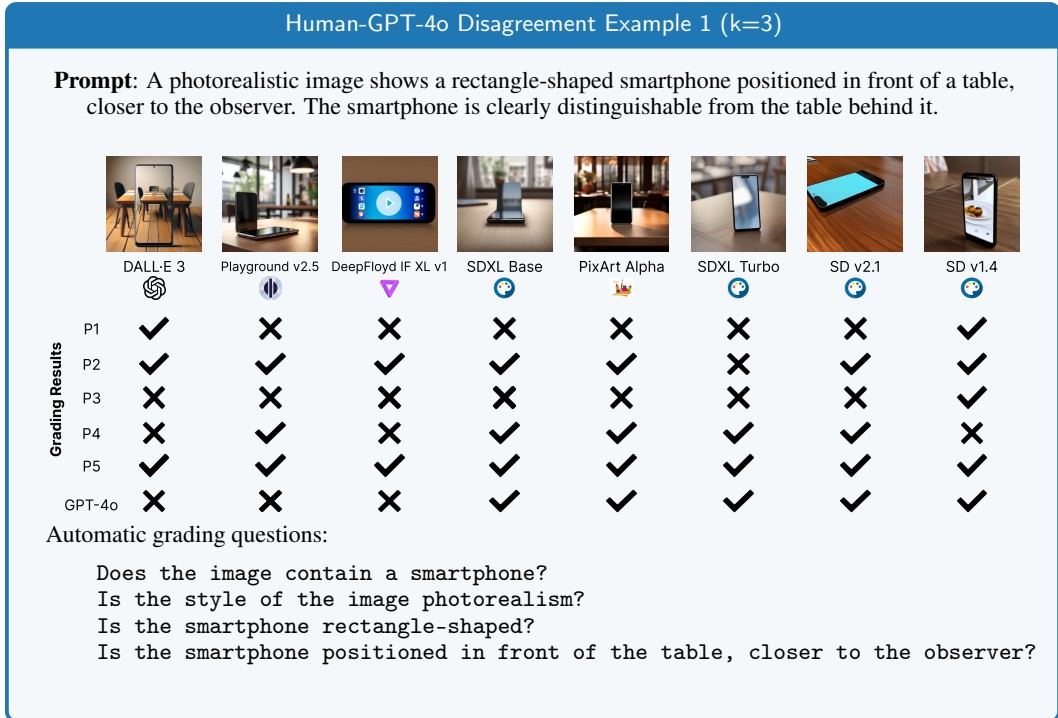

**Human-GPT-4o Disagreement Example 1 (k=3)**

**Prompt**: A photorealistic image shows a rectangle-shaped smartphone positioned in front of a table, closer to the observer. The smartphone is clearly distinguishable from the table behind it.

Automatic grading questions:

```
Does the image contain a smartphone?
Is the style of the image photorealism?
Is the smartphone rectangle-shaped?
Is the smartphone positioned in front of the table, closer to the observer?
```

## Human-GPT-4o Disagreement Example 2 (DALL·E 3, k=4)

**Prompt**: The image shows a red table with a red metallic-textured necklace placed on its surface.

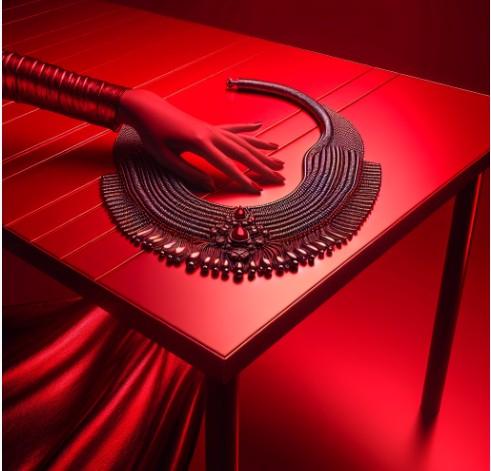

Grading results:

```
Human:
0 1 1 0 1
GPT-4o: 0
```

GPT-4o grading details:

```
Does the image contain a table?          1
Does the image contain a necklace?       1
Is the color of the necklace red?        0
Is the color of the table red?           1
Does the necklace have a metallic texture? 1
```

**Prompt**: A tiny elephant is positioned to the left of a tiny white broccoli.

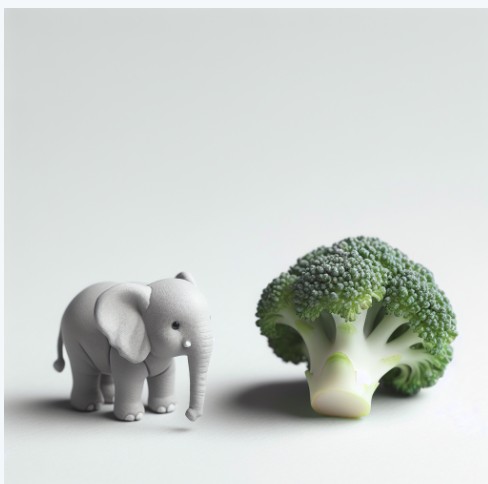

Grading results:

```
Human:
1 0 0 1 1
GPT-4o: 0
```

GPT-4o grading details:

```
Does the image contain an elephant?                        1
Does the image contain a broccoli?                         1
Is the elephant tiny?                                      1
Is the color of the broccoli white?                        0
Is the broccoli tiny?                                      0
Is the elephant positioned on the left side of the broccoli? 1
```

**Prompt**: The image shows a blue robot with a glass texture positioned to the right of a tiny rose. The style of the image is photorealism.

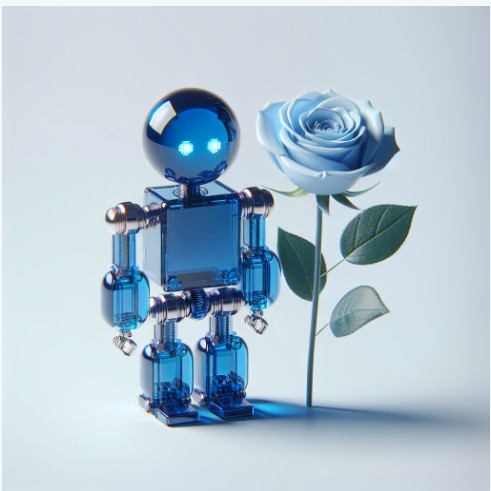

Grading results:

```
Human:
1 0 0 1 1
GPT-4o: 0
```

GPT-4o grading details:

```
Does the image contain a robot?                      1
Does the image contain a rose?                       1
Is the size of the rose tiny?                         0
Is the color of the robot blue?                       1
Is the style of the image photorealism?              0
Does the robot have a glass texture?                  1
Is the robot positioned on the right side of the rose?  0
```

**Prompt**: On a large plate, there is a heart-shaped piece of sushi. Next to it, there is a fork with a glass texture. A tiny butterfly is perched on the edge of the plate. Nearby, a cactus with a fluffy texture is also present.

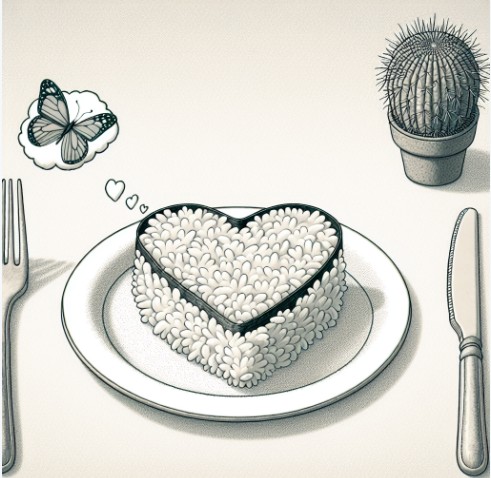

Grading results:

```
Human:
1 0 0 1 1
GPT-4o: 0
```

GPT-4o grading details:

```
Does the image contain a fork?          1
Does the image contain a butterfly?     1
Does the image contain sushi?           1
Does the image contain a cactus?        1
Is the sushi heart-shaped?              1
Does the fork have a glass texture?     0
Is the butterfly tiny?                  0
Does the cactus have a fluffy texture?  0
```

These results highlight the challenges of achieving high inter-human rater reliability in subjective evaluations and show the strengths of our automatic grading method with GPT-4o.

### A.6 Feedback from human evaluators

We received feedback from human evaluators and listed details below.

- There exists phrasing with ambiguity, e.g., in the first example of §A.5, whether it requires the phone to be closer than the front edge of the table, or it covers some part of the table?

- Feedback related to styles: some of the styles are too difficult for models (e.g., expressionism), and some of the styles are difficult to judge (e.g., impressionism); some concepts are hard to realize in certain styles (e.g., "fluffy" texture in "cubism").

- Additional information injected by GPT-4o in prompt generation pipeline: some text prompts contain the quantifier "a single object" even though the individual questions do not require that.

In general, most human evaluators find some images hard to grade and some questions hard to answer, which is aligned with relatively low consistency between human evaluators, observed from Fig. 11. All feedback provides useful insights for future updates of CONCEPTMIX and the development of similar benchmarks. To fairly compensate participants for their time and contributions, each received an hourly wage of $15, with total compensation amounting to $660.

# B  Related Work

## B.1  Compositional Generalization

Compositionality is key to generalizing existing knowledge to new tasks and therefore has attracted significant attention in machine learning. In CV, studies have explored compositional generalization in disentangled representation learning [16, 11, 46], visual relations [27], as well as concept compositions [32]. Other works focus on compositional models for image generation [10], and planning for unseen tasks at inference time [9]. In NLP, compositional generalization has also been studied extensively [13, 23, 4, 20, 21, 28, 30]. SKILL-MIX [47], a more recent evaluation on LLMs, presented a more general approach to evaluate compositional generalization. SKILL-MIX asks LLMs to produce novel pieces of text from random combinations of $k$ skills, which can be made more difficult by simply increasing the value of $k$. CONCEPTMIX is partly inspired by SKILL-MIX, but requires a more complicated design in creating text prompts and effective grading.

## B.2  T2I models and compositional T2I benchmarks

T2I models [35, 2, 3, 5, 33, 43, 25] generate images given text prompts. Traditionally, their performance is evaluated based on alignment with reference (image, caption) pairs. This involves querying the T2I model with the reference caption and assessing the consistency between the generated image and the reference image. Common benchmarks include TIFA160 [18], HRS-Bench [1], DrawBench [37]. When reference images are not provided, benchmarks with prompt templates are used for a more comprehensive measure of compositional capabilities [12, 5, 1, 19, 24]. Among them, the closest to ours is T2I-CompBench [19], which samples complex prompts to evaluate T2I models. However, as noted in Tab. 1, T2I-CompBench limits prompts to 5 concepts, while CONCEPTMIX uses up to 8 (i.e., $k = 7$).

## B.3  Evaluation metrics for generation

Most previous benchmarks use similarity metrics like Inception Score [38, IS], Fréchet Inception Distance [15, FID], and Learned Perceptual Image Patch Similarity [48, LPIPS] to quantify generation quality. These metrics, relying on pre-trained networks, primarily capture pixel-level similarity and often fail to fully capture semantic-level alignment. To address these limitations, recent methods [41, 45, 36] have adopted metrics like CLIPScore [34, 14], which measure cosine similarity between embedded image and text representations, and visual question answering pipelines [22, 49, 26] to better capture text-image alignment. Our evaluation also adopt the visual question answering pipeline for text-image consistency checking, but with a more careful design of asking appropriate questions to verify the generation quality of each visual concept thanks to our prompt generation pipeline.

# C    Benchmark Details

## C.1    Configuration Details

Below are the detailed concept values for each visual concept category in CONCEPTMIX:

**Objects:**  apple, bee, broccoli, butterfly, cactus, car, carrot, cat, chair, chicken, corgi, cow, dirt road, doll, dog, duck, elephant, fork, giraffe, hammer, highway, hill, house, laptop, lion, man, necklace, novel, oak tree, orange, pig, pine tree, pizza, ring, robot, rose, screwdriver, sheep, skyscraper, smartphone, spider, spoon, sunflower, sushi, table, teddy bear, textbook, truck, woman, zebra

**Colors:**  black, blue, brown, gray, green, orange, pink, purple, red, white, yellow

**Numbers:**  2, 3, 4

**Shapes:**  circle, heart, rectangle, square, triangle

**Sizes:**  huge, tiny

**Textures:**  fluffy, glass, metallic

**Spatial Relationship:**  above, behind, below, bottom, in front of, inside, left, outside, right, top

**Styles:**  abstract, cartoon, cubism, expressionism, graffiti, impressionism, ink, manga, oil painting, photorealism, pixel art, pop art, sketch, surrealism, watercolor

Values in blue indicate easy splits, while values in orange denote hard splits of different concepts, as measured on Playground v2.5 with $k = 1$. We use these splits for experiments in §4.3. Note that we use all objects for both easy and hard splits to ensure a fair comparison.

## C.2    Prompt Generation

We use GPT-4o (endpoint of May 13th, 2024), to help bind multiple concepts and generate prompts, as detailed in §4.3. For concept bind, we utilize the JSON format, and start with a JSON in the following structure:

---

**Example of Initial JSON for concept binding**

```
{"objects": [{"id": 1, "item": "teddy bear", "color": "green", "texture":
"glass", "number": "4"}, {"id": 2, "item": "laptop", "shape": "rectangle",
"size": "tiny"}], "style": "oil painting", "relation": [{"name": "behind",
"description": "{ObjectA} is behind {ObjectB}, meaning {ObjectA} is
positioned farther from the observer or camera than {ObjectB}", "ObjectA_id":
"?", "ObjectB_id": "?"}]}
```

---

We intentionally leave some question marks for spatial relationships, and ask GPT-4o to fill them and potentially add new objects if needed. The instruction given to GPT-4o is as follows:

---

**Instructions given to GPT-4o for finalize JSON**

I am trying to create an image containing exactly the following things in a JSON format:
[Initial JSON]
Could you check if there is "?" left in the JSON? If so, could you fill in the missing part? Make sure it makes sense when you fill the missing part. Do not fill in anything else unless it is indicated by "?". You may add additional objects, but only in the following two cases:
* It is needed to fill in any "?" (Note when you fill "?", you should use existing objects first. If you still choose to add an object, explain why the existing objects cannot fulfill the need.); or
* If there is an attribute specified in the JSON that contains relative information (e.g. "size") and there is no other object for reference. (The reason for adding an object for this case is because one cannot tell whether an object is huge without any other object in the image, but we are fine if there is no such attribute mentioned in the JSON. Note other existing objects in JSON can be used for reference, and the reference object does not need to be the same object. If you still choose to add an object, explain why the existing objects cannot fulfill the need.)
DO NOT add any object if none of the above situations is strictly satisfied, and DO NOT try to improve the image in other ways. If you choose to add an object, make sure it fits in the image naturally. Please only add the necessary objects, and the added objects should only have "id" and "item" specified, and should be appended to "objects".

---

After we obtain the final JSON, we use the following instructions to produce text prompts, and we implement a robust prompt rejection mechanism to ensure the reliability of the generated prompts.

---

**Instructions given to GPT-4o for text prompt generation**

Make up a human-annotated description of an image that describe the following properties (meaning you can infer these properties from the description):
[description of properties]
As a reference, I constructed a JSON containing all the information from the properties and some additional information that you should incorporate into your description:
[final JSON]
Describe the image in an objective and unbiased way. Keep the description clear and unambiguous, and synthesize the objects in a clever and clean way, so people can roughly picture the scene from your description. DO NOT introduce unnecessary objects and unnecessary descriptions of the objects beyond the given properties and JSON. If there is an interaction between two objects, make sure the two objects are distinguishable. Avoid any descriptions involving a group of objects, or an ambiguous number of objects like "at least one", "one or more", or "several". Do not add subjective judgments about the image, it should be as factual as possible. Do not use fluffy, poetic language, or any words beyond the elementary school level. Respond "WRONG" and explain if the properties have obvious issues or conflicts, or if it is hard to realize them in an image. Otherwise, respond only with the caption itself.

---

Here the property description of each selected concept category is generated using the template provided in Tab. 6.

Table 6: Template to format selected concepts with their corresponding descriptions presented to GPT-4. Values in brackets [] represent chosen visual concepts from their respective categories.

| Category | Description template |
|---|---|
| Objects | the image contains one or more [object name] |
| Colors | the color of [object name] is [color name] |
| Numbers | the number of [object name] is exactly [number] |
| Shapes | [object name] is [shape name] shaped |
| Sizes | [object name] has a [size value] size |
| Textures | [object name] has a [texture name] texture |
| Spatial, top | [Object A] is on top of [Object B], meaning [Object A] is positioned above or at the highest point of [Object B], touching each other |
| Spatial, bottom | [Object A] is at the bottom of [Object B], meaning [Object A] is positioned below or at the lowest point of [Object B], touching each other |
| Spatial, above | [Object A] is above [Object B], meaning [Object A] is positioned higher than [Object B] without touching it |
| Spatial, below | [Object A] is below [Object B], meaning [Object A] is positioned lower than [Object B] without touching it |
| Spatial, left | [Object A] is positioned on the left side of [Object B] |
| Spatial, right | [Object A] is positioned on the right side of [Object B] |
| Spatial, behind | [Object A] is behind [Object B], meaning [Object A] is positioned farther from the observer or camera than [Object B] |
| Spatial, in front of | [Object A] is on top of [Object B], meaning [Object A] is positioned above or at the highest point of [Object B], touching each other |
| Spatial, inside | [Object A] is inside [Object B], meaning [Object A] is positioned within the boundaries or interior of [Object B] |
| Spatial, outside | [Object A] is outside of [Object B], meaning [Object A] is positioned beyond the boundaries or exterior of [Object B] |
| Styles | the style of the image is [style name] |

**Prompt Rejection Mechanism.** After generating the prompts, we then prompt GPT-4o for validation (see §2.3), using the following instruction:

---

**Instructions given to GPT-4o for prompt validation**

Could you read your caption again and verify if it makes sense in a very loose sense (e.g., a person cannot be triangle-shaped, but a cloud can be square-shaped and a tree can be rectangle-shaped)? If yes, respond with the exact same caption. If not, respond with "WRONG" and explain why.

---

When the system detects a violation of these rules, it responds with "WRONG," providing an explanation for why the prompt is unsuitable. The rejection mechanism is fully automated, ensuring consistency across all generated prompts. Only prompts that meet all criteria are accepted, which improves the reliability of the generated prompts. This resulted in a rejection rate of approximately 13-52% of initially generated prompts, primarily due to the shape information. The rejection rate goes up as k increases. Here are some examples of rejection reasons:

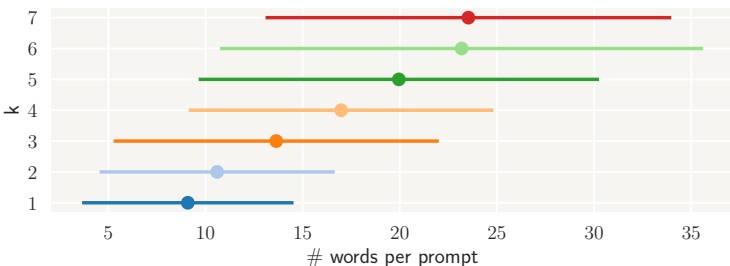

Figure 12: **Prompt Length Distribution.** Larger values of $k$ result in longer and potentially more complex prompts.

- "A triangle-shaped cat is difficult to conceptualize in a realistic image as animals typically do not have geometric shapes."
- "A hill cannot be rectangle-shaped as hills are naturally irregular in shape, and it's not practical to represent them as rectangles in a meaningful context."

**Prompt Length.** We also provide the distribution of text prompt lengths for different values of $k$. The length of the text prompt may indicate the complexity of the task, as longer prompts tend to involve more concepts. The distribution of text prompt lengths for each $k$ is shown in Fig. 12.

**Concept-Prompt Discrepancies.** In examining the discrepancies between the selected concepts and the generated prompts, we find that the generated prompts accurately reflect the visual concepts in the majority of cases. However, in very few instances (approximately 1% of the time), we observe minor discrepancies. For instance:

- Visual Concepts: bee, sushi, cow, man, chair, circle, tiny, cartoon.
- Prompt: In a cartoon-style image, a tiny, circle-shaped cow sits on a chair. A man stands nearby, holding a piece of sushi. A bee is flying above the scene.

In this case, the man "holding a piece of sushi" is not explicitly provided in the selected visual concepts. Nevertheless, the overall high accuracy of concept representation shows the robustness of our prompt generation pipeline, with only minimal refinements potentially needed to capture these rare, more complex scenarios.

### C.3 Question Generation

For each generated prompt, we also accompany it with a list of GPT-4o-generated questions, as detailed in §2.4, which are later used for grading. Specifically, we use the following instruction:

> **Instructions given to GPT-4o for question generation**
>
> A student just draw a picture based on your description. Can you help me verify whether the student did a good job? Specifically, I want to know if the image follows your description and also follows the properties I mentioned earlier. You should ask me one yes or no question for each property, and I will tell you if they are satisfied. For example, for properties like "the image contains one or more [object name]", the corresponding question should be "Does the image contain [object name]". Respond only the $k$ questions, one for each property, in the same order of the properties, and each on a new line.

**Concept-Prompt-Question Discrepancies.** We analyzed 100 randomly sampled prompts to identify cases where concepts mentioned in the prompts were not given in the selected visual concepts. Only one case (1%) showed a mismatch between the prompt and derived questions. For instance:

- Visual Concepts: pine tree, bee, tiny, photorealism, tiny, metallic, top, left
- Prompt: A tiny pine tree on the left side of the image has a tiny metallic bee positioned on top of it. The scene is depicted in a photorealistic style.
- Questions:
  - Does the image contain a pine tree?
  - Does the image contain a bee?
  - Is the pine tree tiny in size?
  - Is the style of the image photorealism?
  - Is the bee tiny in size?

- Does the bee have a metallic texture?
- Is the bee on top of the pine tree?
- Is the pine tree positioned on the left side of the bee?

The final question, `Is the pine tree positioned on the left side of the bee?` inaccurately interprets "left side of the image" as relative to the bee. This single instance of misalignment suggests that the overall concept representations in the prompts and questions are highly accurate, with only minor, isolated discrepancies. In this context, such rare occurrences are negligible and unlikely to significantly impact the evaluation.

# D  Experimental Details

## D.1  T2I Generation Time Cost

All experiments are conducted on a single NVIDIA A6000 GPU card with 48GB memory. Tab. 7 provides statistics on the time cost for each image generation across all the evaluated models.

Table 7: Averaged time cost per generation for evaluated models using a single NVIDIA A6000 GPU card.

| Model | Time cost (seconds) per generation |
|---|---|
| SD v1.4 | 2.17 |
| SDXL Turbo | 0.34 |
| SD v2.1 | 3.99 |
| SDXL Base | 10.03 |
| DeepFloyd IF XL v1 | 18.69 |
| DALL·E 3 | 12.58 |
| Playground v2.5 | 10.17 |
| PixArt alpha | 4.41 |

## D.2  GPT-4o Grading Cost

While open-source model alternatives exist, they currently fall short of GPT-4o's performance, particularly when evaluating complex and compositional image generation tasks. Using less effective models could compromise the quality of the evaluation. Since we only need to generate 300x7 images per model in our current settings, and considering the fact that new image generation models are not released frequently, the overall cost remains feasible within our research budget. Please find detailed cost information on the OpenAI API Pricing webpage[3].

**Input Cost:** 300 (# images) $\times$ 7 (# k) $\times$ 8 (# models) = 16800 images in total.

- **Image:** A 1024x1024 image (the largest size in our experiment) costs $0.003825 using `GPT-4o-2024-05-13`, 16800 $\times$ $0.003825 = $64.26
- **Text:** Each question has roughly 20 words, approximately 27 tokens. With 8 questions per image maximum and 16,800 images, 27 (# tokens) $\times$ 8 (# questions) $\times$ 16,800 (# images) = 3,628,800 tokens, 3,628,800 tokens $\times$ $2.50/1M tokens = $9.07.

**Output Cost:** Assuming each yes/no answer is about 1 token, 8 (# questions) $\times$ 16,800 (# images) = 134,400 tokens, 134,400 (# tokens) $\times$ $7.50/1M tokens = $1.01

**Total:** Roughly $74.34 for our entire grading using GPT-4o across all $k$ and all models.

It is also worth comparing the cost of GPT-4o to that of human evaluation studies. Human studies are significantly more expensive and time-consuming. For context, our human study cost $660 ($15 per person per hour) and required considerable time to organize and conduct. In contrast, using GPT-4o to evaluate a substantial set of images is considerably more cost-effective and can be completed much faster. Moreover, the cost of using the GPT-4o API has been decreasing over time (e.g., $5.00 per 1M input tokens for `GPT-4o-2024-05-13`, but $2.50 per 1M input tokens for `GPT-4o-2024-08-16`), making it an increasingly affordable option.

## D.3  Generation Configurations

For all open-source models, we use their checkpoints from Hugging Face for generation, as listed in Tab. 8, with their default generation configurations. For DALL-E, we generate images via its API endpoint with the default settings[4].

## D.4  Experimental details for §4.5

In §4.5, we analyze the concept diversity of LAION [40] (MIT License). We prompt GPT-4o to identify the number of visual concepts in each sampled caption from LAION:

---

[3]https://openai.com/api/pricing/
[4]https://platform.openai.com/docs/api-reference/images/create

Table 8: Summary of evaluated models with corresponding Hugging Face links and licenses.

| Model | Hugging Face Link |
|---|---|
| SD v1.4 | https://huggingface.co/CompVis/stable-diffusion-v1-4 |
| SDXL Turbo | https://huggingface.co/stabilityai/sdxl-turbo |
| SD v2.1 | https://huggingface.co/stabilityai/stable-diffusion-2-1 |
| SDXL Base | https://huggingface.co/stabilityai/stable-diffusion-xl-base-1.0 |
| DeepFloyd IF XL v1 | https://huggingface.co/DeepFloyd/IF-I-XL-v1.0 |
| Playground v2.5 | https://huggingface.co/playgroundai/playground-v2.5-1024px-aesthetic/ |
| PixArt alpha | https://huggingface.co/PixArt-alpha/PixArt-XL-2-1024-MS |

(a) Models and their Hugging Face links

| Model | License |
|---|---|
| SD v1.4 | CreativeML OpenRAIL M license |
| SDXL Turbo | Stability AI Non-commercial Research Community License |
| SD v2.1 | CreativeML Open RAIL++-M License |
| SDXL Base | CreativeML Open RAIL++-M License |
| DeepFloyd IF XL v1 | DeepFloyd IF License Agreement |
| Playground v2.5 | Playground v2.5 Community License |
| PixArt alpha | CreativeML Open RAIL++-M License |

(b) Models and their licenses

---

**Instructions given to GPT-4o for concept identification**

Given a prompt, identify whether it includes any concept from the following visual concept categories: object, color, number, shape, size, spatial relationship, style, and texture. Directly return the included visual concept categories as your answer. If there is no detected visual concept categories, return an empty string.

---

### D.5   Additional Individual Concept Performance §4.2

Following Fig. 5, we visualize all of the concept categories in Fig. 13.

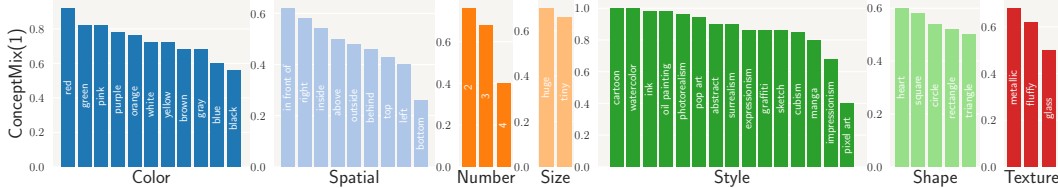

Figure 13: **Performance of concepts within the same category.**

Tab. 9 provides the concept fraction score of all evaluated models, showing a high correlation with the full mark score reported in Tab. 4. Similar to Tab. 4, the concept fraction score drops when increasing $k$, with DALL·E 3 being the best, and SD v1.4 being the worst. Note the drop in concept fraction score not only indicates the difficulty level increase of the whole text prompts but also shows each concept is harder to realize with more concepts described in the prompt.

### D.6   Concept Fraction Score

### D.7   Discussion and Supplementary Grading Experiments

**Discussion on GPT-4 Version Control.** While it is true that GPT will continue to evolve, we believe that this evolution can be an advantage rather than a limitation. As GPT improves its vision understanding capabilities, the correctness evaluation of the generated images will become more accurate, aligning closer to real-world interpretations. This would make future comparisons even more meaningful. Additionally, our primary focus is on the compositional capability of T2I models, more specifically, binding k visual concepts with an object. The consistent evaluation of the compositional capability does not necessarily rely on a static version of GPT but rather on the ability to evaluate increasingly complex and accurate compositions. To verify this, we run additional evaluation experiments with a different VLM and provide the results in the following section.

Table 9: **Performance of T2I Models on our CONCEPTMIX benchmark.** Here we show the concept fraction score with varying difficulty levels $k$ from 1 to 7. As $k$ increases, the performance of all models decreases, but at different rates.

| | $k=1$ | $k=2$ | $k=3$ | $k=4$ | $k=5$ | $k=6$ | $k=7$ |
|---|---|---|---|---|---|---|---|
| SD v1.4 [35] | $0.74_{\pm0.03}$ | $0.61_{\pm0.03}$ | $0.55_{\pm0.03}$ | $0.50_{\pm0.02}$ | $0.44_{\pm0.02}$ | $0.41_{\pm0.02}$ | $0.36_{\pm0.02}$ |
| SD v2.1 [33] | $0.74_{\pm0.03}$ | $0.68_{\pm0.03}$ | $0.61_{\pm0.03}$ | $0.54_{\pm0.03}$ | $0.50_{\pm0.03}$ | $0.48_{\pm0.02}$ | $0.45_{\pm0.02}$ |
| SDXL Turbo [39] | $0.81_{\pm0.03}$ | $0.72_{\pm0.03}$ | $0.65_{\pm0.03}$ | $0.60_{\pm0.03}$ | $0.57_{\pm0.02}$ | $0.54_{\pm0.02}$ | $0.49_{\pm0.02}$ |
| PixArt alpha [7] | $0.82_{\pm0.03}$ | $0.73_{\pm0.03}$ | $0.67_{\pm0.03}$ | $0.61_{\pm0.03}$ | $0.56_{\pm0.02}$ | $0.53_{\pm0.02}$ | $0.49_{\pm0.02}$ |
| SDXL Base [33] | $0.84_{\pm0.03}$ | $0.76_{\pm0.03}$ | $0.69_{\pm0.02}$ | $0.63_{\pm0.02}$ | $0.60_{\pm0.02}$ | $0.57_{\pm0.02}$ | $0.53_{\pm0.02}$ |
| DeepFloyd IF XL v1 [43] | $0.84_{\pm0.03}$ | $0.74_{\pm0.03}$ | $0.66_{\pm0.03}$ | $0.61_{\pm0.02}$ | $0.59_{\pm0.02}$ | $0.55_{\pm0.02}$ | $0.51_{\pm0.02}$ |
| Playground v2.5 [25] | $0.84_{\pm0.03}$ | $0.77_{\pm0.03}$ | $0.71_{\pm0.02}$ | $0.64_{\pm0.02}$ | $0.62_{\pm0.02}$ | $0.58_{\pm0.02}$ | $0.52_{\pm0.02}$ |
| DALL·E 3 [2] | $0.92_{\pm0.02}$ | $0.85_{\pm0.02}$ | $0.83_{\pm0.02}$ | $0.76_{\pm0.02}$ | $0.75_{\pm0.02}$ | $0.72_{\pm0.02}$ | $0.71_{\pm0.02}$ |

**Alternative VLM Evaluations.** We conduct additional grading experiments using different VLMs beyond GPT-4o. We show experimental results with Deepseek-vl-7b-chat [29] in Tab. 10 and Fig. 14 below, and we observe that the relative results and the general trend (performance comparison across different models and different k) still hold ignoring specific VLMs used for grading.

Both models (GPT-4o & Deepseek-vl-7b-chat) consistently rank DALL·E 3 as the top performer across all k values, with SD v1.4 and SD v2.1 performing the worst. DALL·E 3 maintains a significant lead, particularly at $k=3$ (0.62 with Deepseek-vl-7b-chat, 0.50 with GPT-4o). The relative ranking of models remains stable. All models show a clear performance decline as $k$ increases. For instance, DALL·E 3 scores 0.90 at $k=1$ and 0.18 at $k=7$ with Deepseek-vl-7b-chat, while in our GPT-4o results, it scores 0.83 at $k=1$ and 0.08 at $k=7$. Similarly, Playground v2.5 scores 0.81 at $k=1$ and 0.06 at $k=7$ with Deepseek-vl-7b-chat, compared to 0.70 at $k=1$ and 0.01 at $k=7$ with GPT-4o. Notably, Deepseek-vl-7b-chat evaluations show slightly higher numbers than GPT-4o evaluations across the board. We include the visualization comparisons in Fig. 14 to compare the general trends of the two models.

Table 10: **Performance of Eight T2I Models Evaluated Using Deepseek-vl-7b-chat.** We report the full mark scores for different difficulty levels $k$ (1 to 7), representing the proportion of generated images that correctly satisfy all $k+1$ required visual concepts. The results show consistent trends in model performance across difficulty levels, aligning with evaluations using GPT-4o.

| | $k=1$ | $k=2$ | $k=3$ | $k=4$ | $k=5$ | $k=6$ | $k=7$ |
|---|---|---|---|---|---|---|---|
| SD v1.4 [35] | $0.61_{\pm0.06}$ | $0.32_{\pm0.06}$ | $0.15_{\pm0.05}$ | $0.09_{\pm0.04}$ | $0.03_{\pm0.03}$ | $0.02_{\pm0.02}$ | $0.00_{\pm0.01}$ |
| SD v2.1 [33] | $0.64_{\pm0.06}$ | $0.38_{\pm0.06}$ | $0.23_{\pm0.05}$ | $0.14_{\pm0.04}$ | $0.07_{\pm0.04}$ | $0.03_{\pm0.03}$ | $0.02_{\pm0.02}$ |
| SDXL Turbo [39] | $0.74_{\pm0.05}$ | $0.53_{\pm0.06}$ | $0.32_{\pm0.06}$ | $0.17_{\pm0.05}$ | $0.09_{\pm0.04}$ | $0.05_{\pm0.03}$ | $0.03_{\pm0.03}$ |
| PixArt alpha [7] | $0.76_{\pm0.05}$ | $0.48_{\pm0.06}$ | $0.38_{\pm0.06}$ | $0.20_{\pm0.05}$ | $0.11_{\pm0.04}$ | $0.08_{\pm0.04}$ | $0.04_{\pm0.03}$ |
| DeepFloyd IF XL v1 [43] | $0.73_{\pm0.05}$ | $0.48_{\pm0.06}$ | $0.31_{\pm0.06}$ | $0.17_{\pm0.05}$ | $0.10_{\pm0.04}$ | $0.05_{\pm0.03}$ | $0.01_{\pm0.02}$ |
| SDXL Base [33] | $0.74_{\pm0.05}$ | $0.54_{\pm0.06}$ | $0.27_{\pm0.05}$ | $0.16_{\pm0.05}$ | $0.12_{\pm0.04}$ | $0.05_{\pm0.03}$ | $0.03_{\pm0.03}$ |
| Playground v2.5 [25] | $0.81_{\pm0.05}$ | $0.59_{\pm0.06}$ | $0.41_{\pm0.06}$ | $0.20_{\pm0.05}$ | $0.14_{\pm0.04}$ | $0.08_{\pm0.04}$ | $0.06_{\pm0.03}$ |
| DALL·E 3 [2] | $0.90_{\pm0.04}$ | $0.74_{\pm0.05}$ | $0.62_{\pm0.06}$ | $0.41_{\pm0.06}$ | $0.33_{\pm0.06}$ | $0.28_{\pm0.05}$ | $0.18_{\pm0.05}$ |

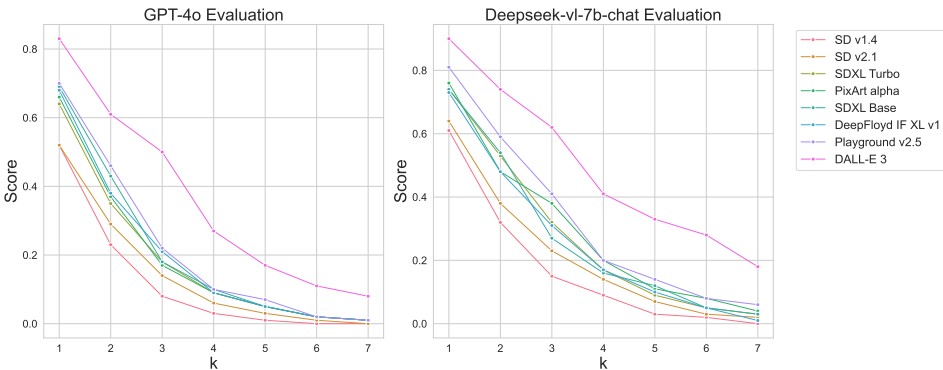

Figure 14: **Comparison of Different Grading Models.** Here we show the comparisons of image generation model performance evaluated by GPT-4o (left) and Deepseek-vl-7b-chat (right) across different k values. Both evaluations consistently rank DALL·E 3 as the top performer, with SD v1.4 and SD v2.1 performing the worst. All models show a clear performance decline as k increases. The relative ranking of models remains stable across both evaluations, though Deepseek-vl-7b-chat tends to assign slightly higher scores overall compared to GPT-4o.

# E  Common Failure Cases

In this section, we analyze frequent failure cases faced by T2I models, and we provide the visualizations of two failure cases across all visual concept categories.

## E.1  Numbers

**Numbers Failure Case (Example 1, Playground v2.5)**

**Prompt**: The image shows four elephants and one zebra standing on a grassy plain.

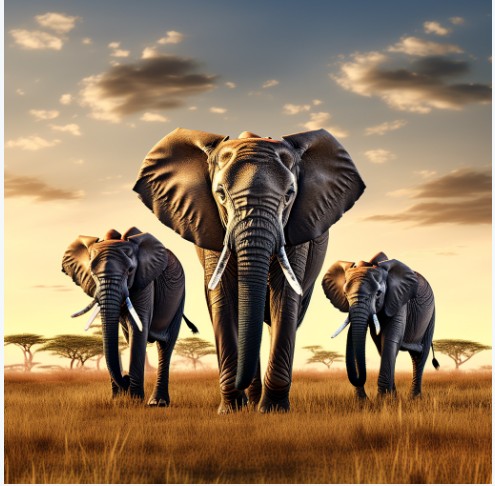

**Prompt Generation:**
```
{
    "num_skills": 2,
    "categories": ["object", "object", "number"],
    "visual_concepts": ["elephant", "zebra", "4"]
}
```

**Grading Results:**
```
{
"questions": [
    "Does the image contain elephants?  ",
    "Does the image contain zebras?  ",
    "Does the image contain exactly 4 elephants?"
    ],
"scores": [
    1,
    0,
    0
    ]
}
```

**Prompt**: In a pop art style image, there are two huge glass-textured carrots. In front of the carrots, there are three tiny giraffes. Additionally, there is an apple included in the scene.

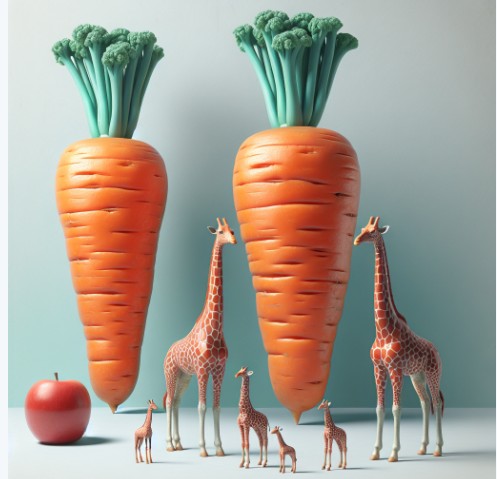

**Prompt Generation:**

```
{
    "num_skills": 7,
    "categories": [
        "object", "object", "number", "size", "number", "texture",
        "style", "size"
    ],
    "visual_concepts": [
        "carrot", "giraffe", "3", "tiny", "2", "glass", "pop art", "huge"
    ]
}
```

**Grading Results:**

```
{
    "questions": [
        "Does the image contain one or more carrots?  ",
        "Does the image contain one or more giraffes?  ",
        "Does the image contain exactly 3 giraffes?  ",
        "Are the giraffes tiny in size?  ",
        "Does the image contain exactly 2 carrots?  ",
        "Do the carrots have a glass texture?  ",
        "Is the style of the image pop art?  ",
        "Are the carrots huge in size?"
    ],
    "scores": [
        1,
        1,
        0,
        1,
        1,
        0,
        0,
        1
    ]
}
```

**E.2  Shapes**

**Prompt**: A tiny yellow sheep stands on a heart-shaped highway. Nearby, a small corgi sits next to a piece of sushi.

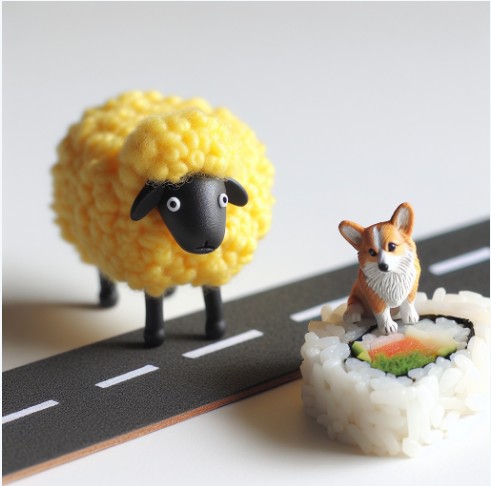

**Prompt Generation:**

```
{
    "num_skills": 7,
    "categories": [
        "object", "object", "object", "object", "shape", "color",
        "size", "size"
    ],
    "visual_concepts": [
        "sheep", "highway", "sushi", "corgi", "heart", "yellow",
        "tiny", "tiny"
    ]
}
```

**Grading Results:**

```
{
    "questions": [
        "Does the image contain sheep?  ",
        "Does the image contain a highway?  ",
        "Does the image contain sushi?  ",
        "Does the image contain a corgi?  ",
        "Is the highway heart-shaped?  ",
        "Is the color of the sheep yellow?  ",
        "Is the sheep tiny in size?  ",
        "Is the corgi tiny in size?"
    ],
    "scores": [
        1,
        0,
        1,
        1,
        0,
        1,
        1,
        1
    ]
}
```

**Prompt**: A huge, white, heart-shaped table is placed next to a chair.

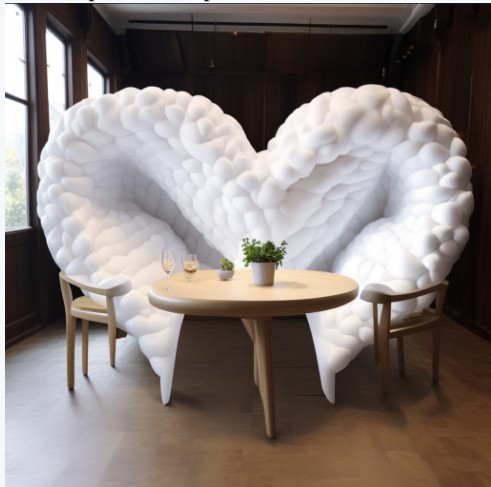

**Prompt Generation:**

```
{
    "num_skills": 3,
    "categories": ["object", "size", "color", "shape"],
    "visual_concepts": ["table", "huge", "white", "heart"]
}
```

**Grading Results:**

```
{
    "questions": [
        "Does the image contain a table?  ",
        "Is the table huge in size?  ",
        "Is the color of the table white?  ",
        "Is the shape of the table heart-shaped?"
    ],
    "scores": [
        1,
        0,
        0,
        0
    ]
}
```

### E.3 Sizes

**Prompt**: In an oil painting, a tiny corgi is positioned in front of three tiny brown volcanoes.

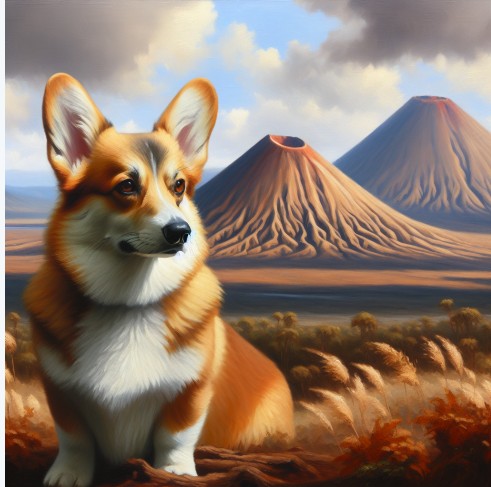

**Prompt Generation:**

```
{
    "num_skills": 7,
    "categories": [
        "object", "object", "color", "style",
        "size", "number", "size", "spatial"
    ],
    "visual_concepts": [
        "corgi", "volcano", "brown", "oil painting", "tiny", "3",
        "tiny", "in front of"
    ]
}
```

**Grading Results:**

```
{
    "questions": [
        "Does the image contain corgi?",
        "Does the image contain volcano?",
        "Is the color of the volcano brown?",
        "Is the style of the image oil painting?",
        "Is the size of the volcano tiny?",
        "Is the number of volcanoes exactly 3?",
        "Is the size of the corgi tiny?",
        "Is the corgi positioned in front of the volcano?"
    ],
    "scores": [
        1,
        1,
        1,
        1,
        0,
        0,
        0,
        1
    ]
}
```

**Prompt**: In an oil painting, a huge smartphone rests on a table next to a green corgi. A tiny hammer with a fluffy texture is also on the table, alongside a book.

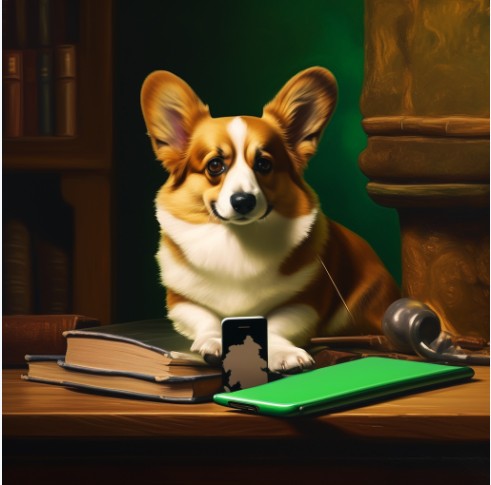

**Prompt Generation:**

```
{
"num_skills": 7,
"categories": [
    "object", "object", "object", "size", "texture",
    "color", "size", "style"
],
"visual_concepts": [
    "smartphone", "corgi", "hammer", "huge", "fluffy",
    "green", "tiny", "oil painting"
]
}
```

**Grading Results:**

```
{
"questions": [
    "Does the image contain a smartphone?",
    "Does the image contain a corgi?",
    "Does the image contain a hammer?",
    "Is the smartphone huge in size?",
    "Is the hammer fluffy in texture?",
    "Is the corgi green in color?",
    "Is the hammer tiny in size?",
    "Is the style of the image oil painting?"
],
"scores": [
    1,
    1,
    1,
    0,
    0,
    0,
    1,
    1
]
}
```

## E.4 Textures

**Prompt**: A scene shows a glass-textured laptop on a desk beside a glass-textured robot. In the background, there is a duck standing on the floor next to a cactus.

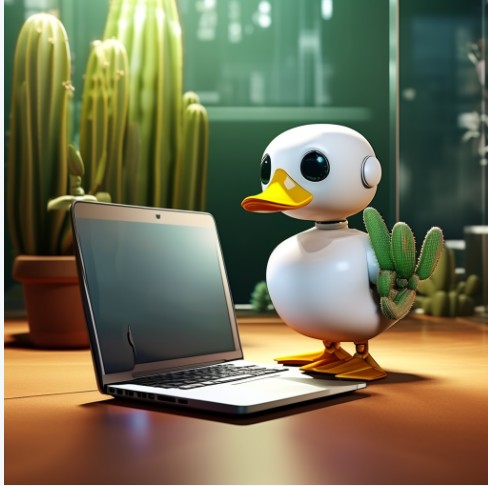

**Prompt Generation:**

```
{
    "num_skills": 5,
    "categories": [
        "object", "object", "object", "object", "texture", "texture"
    ],
    "visual_concepts": [
        "laptop", "robot", "duck", "cactus", "glass", "glass"
    ]
}
```

**Grading Results:**

```
{
    "questions": [
        "Does the image contain a laptop?  ",
        "Does the image contain a robot?  ",
        "Does the image contain a duck?  ",
        "Does the image contain a cactus?  ",
        "Does the robot have a glass texture?  ",
        "Does the laptop have a glass texture?"
    ],
    "scores": [
        1,
        1,
        1,
        1,
        0,
        1
    ]
}
```

**Prompt**: In a vibrant countryside scene, a single wooden house stands in a field. Nearby, a corgi with a short tail observes a sheep grazing on the lush, green grass. In the background, a fluffy-textured volcano looms under a clear blue sky. On a wooden bench beside the house, a yellow screwdriver lies next to a metal hammer.

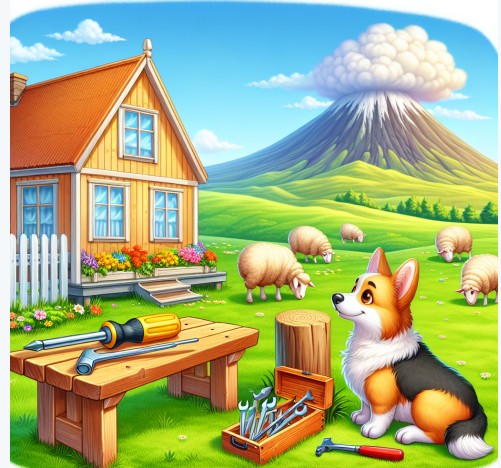

**Prompt Generation:**

```
{
    "num_skills": 7,
    "categories": [
        "object", "object", "object", "object", "object",
        "object", "color", "texture"
    ],
    "visual_concepts": [
        "house", "corgi", "sheep", "volcano", "screwdriver", "hammer",
        "yellow", "fluffy"
    ]
}
```

**Grading Results:**

```
{
    "questions": [
        "Does the image contain a house?  ",
        "Does the image contain a corgi?  ",
        "Does the image contain a sheep?  ",
        "Does the image contain a volcano?  ",
        "Does the image contain a screwdriver?  ",
        "Does the image contain a hammer?  ",
        "Is the color of the screwdriver yellow?  ",
        "Does the volcano have a fluffy texture?"
    ],
    "scores": [
        1,
        1,
        1,
        1,
        1,
        1,
        1,
        0
    ]
}
```

## E.5 Spatial Relationship

---

**Spatial Failure Case (Example 1, DeepFloyd IF XL v1)**

**Prompt**: A tiny glass-textured duck is positioned on the right side of a rock in an ink-style image.

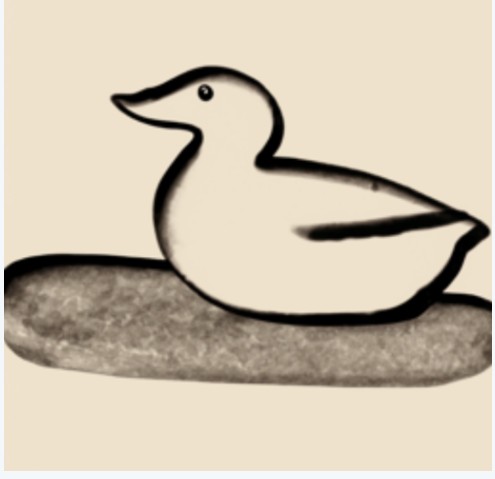

**Prompt Generation:**

```
{
    "num_skills": 4,
    "categories": [
        "object", "size", "texture", "style", "spatial"
    ],
    : [
        "duck", "tiny", "glass", "ink", "right"
    ]
}
```

**Grading Results:**

```
{
    "questions": [
        "Does the image contain a duck?",
        "Is the size of the duck tiny?",
        "Does the duck have a glass texture?",
        "Is the style of the image ink?",
        "Is the duck positioned on the right side of the rock?"
    ],
    "scores": [
        1,
        0,
        0,
        1,
        0
    ]
}
```

---

**Prompt**: The image shows four white, triangle-shaped pine trees with a fluffy texture. A rock is positioned at the bottom of each pine tree, touching them.

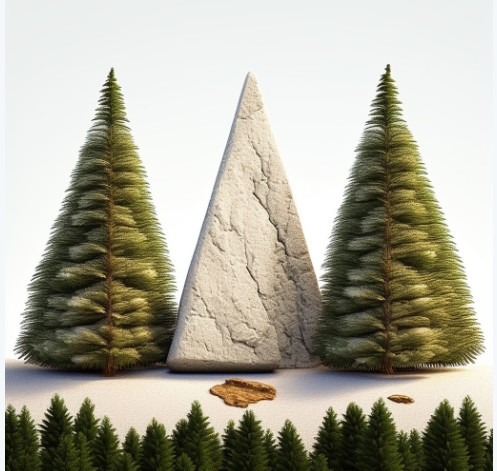

**Prompt Generation:**

```
{
    "num_skills": 5,
    "categories": [
        "object", "shape", "color", "texture", "number", "spatial"
    ],
    : [
        "pine tree", "triangle", "white", "fluffy", "4", "bottom"
    ]
}
```

**Grading Results:**

```
{
    "questions": [
        "Does the image contain pine trees?  ",
        "Are the pine trees triangle shaped?  ",
        "Are the pine trees white in color?  ",
        "Do the pine trees have a fluffy texture?  ",
        "Is the number of pine trees exactly four?  ",
        "Is a rock positioned at the bottom of each pine tree,
        touching them?"
    ],
    "scores": [
        1,
        1,
        0,
        1,
        0,
        0
    ]
}
```

**E.6 Styles**

**Prompt**: A brown duck in an expressionist style.

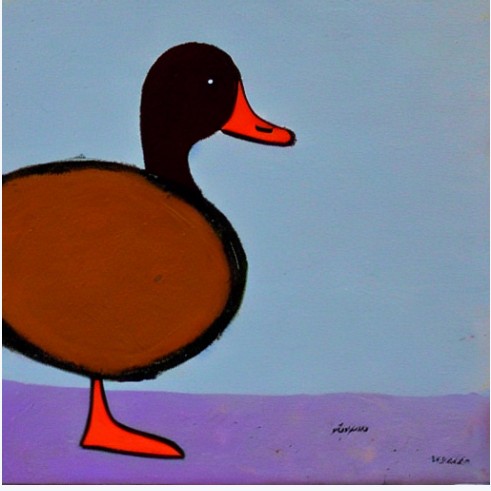

**Prompt Generation:**
```
{
    "num_skills": 2,
    "categories": ["object", "color", "style"],
    "visual_concepts": ["duck", "brown", "expressionism"],
    "question": [
        "Does the image contain a duck?  ",
        "Is the duck brown?  ",
        "Is the style of the image expressionism?"
    ]
}
```

**Grading Results:**
```
{
    "questions": [
        "Does the image contain a duck?  ",
        "Is the duck brown?  ",
        "Is the style of the image expressionism?"
    ],
    "scores": [
        1,
        1,
        0
    ]
}
```

**Prompt**: A huge fork is positioned nearer to the observer than a plate in an impressionism-style image.

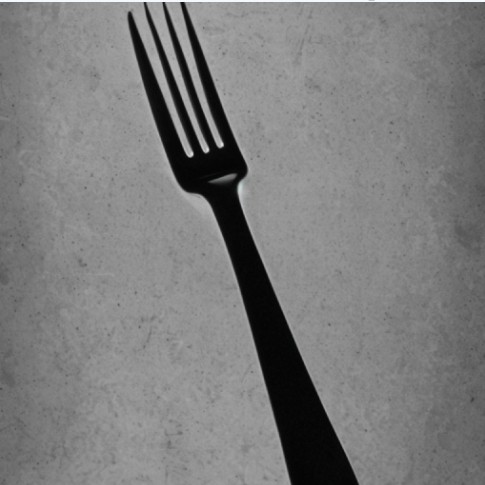

**Prompt Generation:**

```
{
    "num_skills": 3,
    "categories": [
        "object", "style", "size", "spatial"
    ],
    "visual_concepts": [
        "fork", "impressionism", "huge", "in front of"
    ]
}
```

**Grading Results:**

```
{
    "questions": [
        "Does the image contain a fork?",
        "Is the style of the image impressionism?",
        "Is the fork huge?",
        "Is the fork positioned nearer to the observer or camera than
        the plate?"
    ],
    "scores": [
        1,
        0,
        0,
        0
    ]
}
```

## E.7 Colors

**Prompt**: The image shows a green cow standing beside a tiny truck. There is a hammer placed on the ground near them, and a large bicycle is parked in the background.

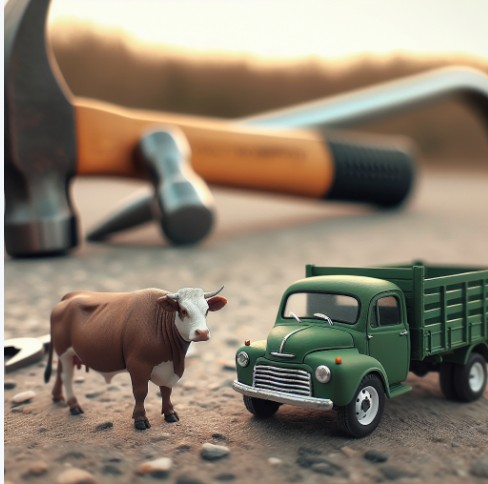

**Prompt Generation:**

```
{
    "num_skills": 4,
    "categories": [
        "object", "object", "object", "size", "color"
    ],
    "visual_concepts": [
        "hammer", "truck", "cow", "tiny", "green"
    ]
}
```

**Grading Results:**

```
{
    "questions": [
        "Does the image contain a hammer?  ",
        "Does the image contain a truck?  ",
        "Does the image contain a cow?  ",
        "Is the truck tiny?  ",
        "Is the cow green?"
    ],
    "scores": [
        1,
        1,
        1,
        1,
        0
    ]
}
```

## Colors Failure Case (Example 2, PixArt alpha)

**Prompt**: The graffiti-style image features a gray cat and a zebra.

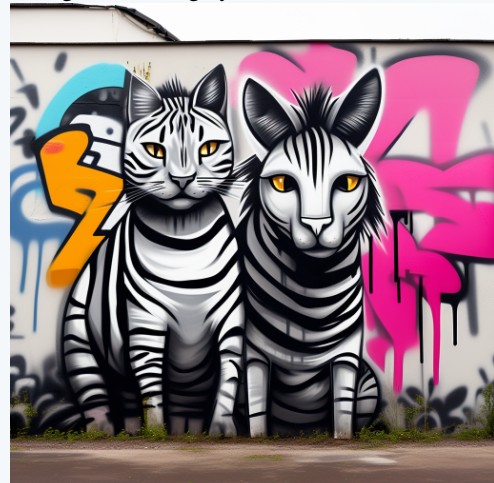

**Prompt Generation:**

```
{
    "num_skills": 3,
    "categories": [
        "object", "object", "color", "style"
    ],
    "visual_concepts": [
        "zebra", "cat", "gray", "graffiti"
    ]
}
```

**Grading Results:**

```
{
"questions": [
    "Does the image contain a zebra?",
    "Does the image contain a cat?",
    "Is the color of the cat gray?",
    "Is the style of the image graffiti?"
],
"scores": [
    0,
    1,
    0,
    1
]
}
```

