# OpenReview forum: "ConceptMix: A Compositional Image Generation Benchmark with Controllable Difficulty"
_NeurIPS.cc/2024/Datasets_and_Benchmarks_Track — NeurIPS 2024 Track Datasets and Benchmarks Poster_

### Official Review · Reviewer_a4xv · 2024-07-22

**Rating:** 8
**Confidence:** 4
**Correctness:** The method is sound and seems reliable.
**Clarity:** The work is clear and easy to follow.

**Review:**

The ConceptMix benchmark builds on previous works that introduce benchmarks for measuring consistency of generated images to input prompts in text-to-image models. This work is distinguished from previous works in its coverage across many concepts, flexibility to expansion and increasingly complex prompts, and additional discriminative power. In addition, the authors employ the benchmark in a useful analysis of existing state of the art models and perform analyses within concept categories among easy/hard variations in given categories. Furthermore, they show the improved discriminative power of the benchmark over the existing T2I-CompBench. Finally, the authors provide a useful early analysis of the LAION training dataset using the evaluation.

The work is well-written and easy to follow. The main improvement areas pertain to additional details about why certain decisions were made, guidance for deployment, and possible additional ablations of the benchmarking method. Please see "Opportunities for improvement"

**Strengths:**

* The benchmark uses careful consideration of the composability of certain concepts and filters out concepts that may lead to unreliable results, such as subjective attributes, nested concepts, or especially difficult concpets for diffusion models.
* The method of building prompts from a set of decomposed concepts also helps in the generation of more reliable QA pairs for evaluation
* The work contains interesting, thorough, and new analyses that are enabled by benchmark, such as study of easy/hard concepts within a single category and the role of a large number of concepts in a given prompt
* The meta evaluation comparing ConceptMix to T2I-CompBench is useful in demonstrating the unique value of ConceptMix
* The paper is well written and easy to follow.

**Additional Feedback:**

N/A

**Documentation:**

Yes, there is good detail about the data collection and organization, availability and maintenance, and ethical and responsible use. In addition, the examples of failure cases and analysis an relation to human perception is very useful.

**Limitations:**

The main limitations of this work relate to the use of automatic question generation and benchmarking methods, which contain some uncertainty. This is discussed in Section 5.

**Opportunities For Improvement:**

* The work would be strengthened with additional detail about the "prompt rejection mechanism" (L 126-7), as this is useful for ensuring that the benchmark is reliable.
* There could be additional study of the "Concept Evaluation" step's reliability. For example, does using a different method of creating the yes/no questions (alternative VQA model or Davidsonian Scene Graph) change the questions that are created? Is the performance of GPT-4o in answering questions consistent across k values or does it struggle when there are many concepts in the image?
* Nit: the comment about ensuring "information injection is minimal and natural at each step" can be more clear and specific.

**Relation To Prior Work:**

The work builds usefully on existing work.

**Summary And Contributions:**

This work introduces ConceptMix, which is a composable benchmark for measuring consistency of text-to-image models. The benchmark contains different categories of visual concepts (both objects and descriptors like color/spatial relationships) and can extend to a variable set of concepts per prompt. To perform evaluations, the benchmark uses an LLM to generate one question per concept to enable decomposed evaluations. The authors use ConceptMix to audit existing models and find that it provides additional insights over existing methods.

---

> ### Author Rebuttal · Authors · 2024-08-16
>
> Thank you very much for the positive feedback! We appreciate the insightful feedback and the recognition of our approach is highly encouraging. We address the questions raised below.
>
> 1. **Details on prompt rejection mechanism:**
>
> More details about the prompt rejection mechanism are in Appendix Sec. C.2. Our prompt for prompt generation explicitly prohibits unnecessary objects, vague quantifiers, and subjective judgments while emphasizing simplicity, clarity, and objectivity. The full prompt reads: “..Respond “WRONG” and explain if the properties have obvious issues or conflicts, or if it is hard to realize them in an image. Otherwise, respond only with the caption itself.”
>
> This resulted in a rejection rate of approximately 13-52% of initially generated prompts, primarily due to the shape information. The rejection rate goes up as k increases. Here are some examples for rejection reasons:
>
> - "A triangle-shaped cat is difficult to conceptualize in a realistic image as animals typically do not have geometric shapes.”
> - “A hill cannot be rectangle-shaped as hills are naturally irregular in shape, and it's not practical to represent them as rectangles in a meaningful context.”
>
>  When the system detects a violation of these rules, it responds with "WRONG," providing an explanation for why the prompt is unsuitable. The rejection mechanism is fully automated, ensuring consistency across all generated prompts. Only prompts that meet all criteria are accepted, which improves the reliability of the generated prompts. We may not be able to fit in these results into the main paper, but we will add an explicit pointer to Appendix C.2.
>
> 2. **Additional study of the reliability of Concept Evaluation step**
>
> A. Regarding the yes/no question generation, our analysis shows that the questions generated have very minimal variations across different methods (more details about question generation are included in Appendix Sec. C.3). The consistency in question generation is largely due to the structured nature of our visual concepts and the careful design of our prompts.
>
> To further validate this consistency, we conducted additional experiments using Claude 3.5 Sonnet for question generation. The results show that the questions generated by Claude 3.5 Sonnet were essentially identical in meaning with some variations in phrasing to those generated by our primary method.
>
> For example when we select the following visual concepts
>
> "concepts": [ "smartphone", "metallic", "3", "brown", "impressionism"]
>
> from those categories:
>
> "categories": [ "object", "texture", "number", "color", "style" ]
>
> The generated questions using Claude are:
>
> - "question": [ "Does the image contain smartphones?", "Do the smartphones have a metallic texture?", "Are there exactly 3 smartphones in the image?", "Are the smartphones brown in color?", "Is the style of the image impressionism?" ]
>
> The generated questions using GPT-4o are:
> - "question": [ "Does the image contain one or more smartphones?", "Does the smartphone have a metallic texture?", "Is the number of smartphones in the image exactly 3?", "Is the color of the smartphone brown?", "Is the style of the image impressionism?" ]
>
> B. To address the reviewer's concerns about reliability, we have added further analysis in the paper and refer them to our reply in the **general response (3)**.
>
> C. Regarding the reviewer's concern about GPT-4o's consistency in answering questions across different k values and its potential difficulty with images containing multiple concepts, we further conducted a detailed analysis across various k values for DALL·E 3. We further provide the consistency map in the **attached pdf file (Fig.1)**.
>
>
> | k | Consistency |
> |---|---------------------|
> | 1 | 0.96 |
> | 2 | 0.84 |
> | 3 | 0.80 |
> | 4 | 0.76 |
> | 5 | 0.64 |
> | 6 | 0.72 |
> | 7 | 0.92 |
>
> The consistency generally decreases as k increases, with a dip observed at k=5 (64%), reflecting the increasing complexity of the tasks. Interestingly, there is a noticeable rebound in consistency at k=6 (72%) and k=7 (92%), which might be attributed to the increasing complexity of compositional generation as k grows. As the task becomes more challenging, the probability of generating fully correct images approaches zero. Consequently, both GPT-4o and human evaluators might converge on similar evaluation. Overall, these findings suggest that GPT-4o is capable of maintaining strong performance even as the complexity of the task varies, although some variability is observed in mid-range k values.
>
>
> 3. **Information injection details.**
>
> We appreciate the reviewer's feedback on clarifying our statement about information injection. To address this, we will revise the text to be more specific and include a detailed explanation. We will add the following analysis to the paper:
>
> In examining the information injection between the selected concepts and the generated sentences, we find that the generated sentences accurately reflect the visual concepts in the majority of cases. However, in very few instances (approximately 1% of the time), we observe minor information injection. For instance:
>
> Visual concepts: bee, sushi, cow, man, chair, circle, tiny, cartoon.
>
> Prompt: In a cartoon-style image, a tiny, circle-shaped cow sits on a chair. A man stands nearby, holding a piece of sushi. A bee is flying above the scene.
>
> In this case, the man "holding a piece of sushi" is not explicitly provided in the selected visual concepts. Nevertheless, the overall high accuracy of concept representation shows the robustness of our prompt generation pipeline, with only minimal refinements potentially needed to capture these rare, more complex scenarios.

---

> > ### Comment · Reviewer_a4xv · 2024-08-23
> >
> > Thanks for the detailed consideration of my review. I appreciate the thoughtful responses, and find the study about consistency of GPT-4o across k-values quite interesting.

---

> > > ### Author Rebuttal · Authors · 2024-08-28
> > >
> > > We really appreciate your positive comments! Thanks for recognizing our work and please don’t hesitate to leave a comment if you have further questions!

---

### Official Review · Reviewer_r238 · 2024-07-24
**clarity issues, potential biases, and evaluation methodology concerns in conceptmix benchmark for t2i models**

**Rating:** 6
**Confidence:** 4

**Review:**

#### General
The extensive use of footnotes makes the reading difficult. Consider embedding this information directly into the text.

#### 2.2 Selecting Visual Concepts
Filtering concepts such as "spongy" texture, challenging items, or those difficult to judge objectively, may cause bias. It might be better to filter these concepts in later stages to avoid excluding potentially valuable examples early, thereby enriching the dataset.

#### 2.3 Compositional Prompt Generation
Using GPT-4o to generate and validate prompts can introduce bias, as it may be predisposed to favor its own outputs. Testing other models as validators/judges, such as Gemini or LLava, would help justify this choice. If this has been done, the results should be elaborated.

#### 2.4 Concept Evaluation
The same consideration applies to concept evaluation. If other models have been tested as judges/validators, those results should be included. If concept evaluation questions were created automatically, it should be ensured that they effectively capture the concepts, or this should be noted as a limitation.

#### 3 Experiments
The results indicate that DALL-E 3 outperforms other models, but GPT-4o is the only judge used. Since these models are from the same company and might share data, using GPT-4o alone may not be sufficient. Including evaluations from other models or human annotations would provide a more balanced assessment.

Human evaluation details provided in Appendix A should be included in the main paper, as they are an integral part of the work. Excluding this section from the main paper raises questions about the validity of the evaluation. It should be mentioned appropriately in the paper, potentially even in the introduction.

**Strengths:**

The automatic process used to create challenging data effectively generates complex T2I data at scale. The images are well-crafted, and the analyses are insightful.

**Additional Feedback:**

See the review, and please include the link to the project's GitHub repository in the main text of the paper.

**Clarity:**

The paper is generally well written, but there are crucial parts missing from the main text that affect clarity. Including all relevant details in the main paper, rather than relying heavily on footnotes and appendices, will improve readability and comprehension. Additionally, integrating human evaluation details into the main text will provide a more comprehensive understanding of the work.

**Correctness:**

Please see my review for detailed comments. You have used only one automatic judge; consider including additional VLMs as judges to enhance the evaluation. Also, incorporate the human evaluation section into the main paper.

**Documentation:**

Yes, the authors have provided sufficient detail on data collection, organization, availability, and maintenance. The submission includes documentation and intended uses, a URL for reviewer access to the dataset, and a hosting, licensing, and maintenance plan. For the benchmark, there is enough detail to support reproducibility.

**Ethics:**

The data is automatically generated for T2I models, so there are no significant ethical considerations, except for ensuring the licensing of GPT outputs is properly addressed.

**Limitations:**

Yes.

**Opportunities For Improvement:**

The writing is generally acceptable, but there are crucial parts missing from the main paper. Please refer to my detailed review for specific areas that need attention. Additionally, include the link to the project's GitHub repository in the main text of the paper, not just in the appendix.

**Relation To Prior Work:**

Yes, the authors have clearly discussed how this work differs from previous contributions. They have effectively highlighted the novel aspects of their benchmark and how it improves upon existing works.

**Summary And Contributions:**

ConceptMix is a new benchmark designed to evaluate T2I models by combining multiple visual concepts in prompts. It addresses limitations in existing benchmarks by dynamically generating diverse prompts involving eight categories of visual concepts. Results indicate performance challenges for T2I models as prompt complexity increases (higher k values), underscoring the need for improved training data and benchmarks to enhance compositional generation capabilities.

---

> ### Author Rebuttal · Authors · 2024-08-16
>
> We sincerely appreciate your constructive and very thorough feedback. We address the points raised in your review below.
>
> 1. **Writing**:
>
> To improve readability, we have integrated the majority of the footnote content directly into the main paper, keeping only 2 footnotes (previously numbered 3 and 6). Additionally, we have moved the human studies from the appendix to the main body of the paper and highlighted the findings in the introduction. Additionally, we also moved the github link to the main paper.
>
> 2. **Selecting Visual Concepts**:
>
> Our current approach of filtering concepts like “spongy” texture or those difficult to judge objectively was designed to prioritize consistency and reliability of our benchmark, and to build a solid foundation of clearly definable visual concepts. However, we acknowledge the potential value in including a broader spectrum of concepts.  In future iterations of our work, we will consider incorporating these more challenging concepts, especially if the VLMs (i.e. GPT-4o in our case) used for auto-gradings continue to advance in their capabilities.
>
> 3. **Testing other models as validators/judges**: (Compositional Prompt Generation & Experiments)
>
> Here we use deepseek-vl-7b-chat[1] as an alternative auto-grader, and we kindly refer the reviewers to the results provided in the **general response (2.)**. Both evaluations (GPT-4o & Deepseek-vl) consistently rank DALL·E 3 as the top performer across all k values, with SD v1.4  and SD v2.1 performing the worst. DALL·E 3 maintains a significant lead, particularly at k=3 (0.62 with deepseek-vl, 0.50 with GPT-4o). The relative ranking of models remains stable. All models show a clear performance decline as k increases.
>
> [1] Lu, Haoyu, et al. "Deepseek-vl: towards real-world vision-language understanding." arXiv 2024.
>
> 4. **Concept Evaluation.**
>
> “If concept evaluation questions were created automatically, it should be ensured that they effectively capture the concepts, or this should be noted as a limitation.” We kindly refer the reviewers to the reply we provided in the **general response (3.)**.

---

> > ### Comment · Reviewer_r238 · 2024-08-25
> >
> > Thank you for addressing my concerns and updating the paper. I still believe adding more auto-graders/judges, especially more familiar vision and language models as previously mentioned, would improve the paper. I will maintain my initial score (6: Marginally above acceptance threshold).

---

> > > ### Author Rebuttal · Authors · 2024-08-29
> > >
> > > Thank you for your feedback again and we really appreciate your suggestion! To address this concern, we've included the additional auto-graders/judges experiments with DeepSeek (see general response 2.) in the appendix of the paper. This experiment provides further comparisons with additional VLMs. We plan to include more evaluation results with additional auto-graders in future work. Thank you again for your thoughtful review!

---

### Official Review · Reviewer_UaRX · 2024-07-25

**Rating:** 5
**Confidence:** 5
**Correctness:** The claims made in the submission are…
**Clarity:** The paper is well written and easy to…

**Review:**

The paper proposes ConceptMix, a compositional T2I generation benchmark. The benchmark demonstrates more diversity in text prompt generation with different visual concepts, and discriminate evaluation results among T2I generation models.
Pros:
1.	The paper proposes a novel benchmark, ConceptMix, which introduces a more diverse and complex text prompts into the evaluation of compositionality for text-to-image generation models.
2.	The paper provides detailed definition and combination of different visual concepts for enlarging compositionality benchmark, which is a good supplementary for existing compositional T2I benchmarks. It offers insight for a scalable and customizable solution for evaluating the compositionality.
3.	The benchmark utilizes discriminative metrics that identify differences in T2I generation models' compositionality. This helps to distinguish the performances of different T2I generation models with increased visual concepts.

Cons
1.	My biggest concern is that the proposed evaluation metric relies on GPT-4o for generating and grading prompts. However, as GPT-4o keeps evolving, it cannot serve as a consistent and fair evaluation metric. Ordinary users do not have version control of GPT-4o models. If in the future, GPT-4o evolves and the model parameters are updated, we cannot access previous versions of GPT-4o, which makes it difficult to conduct fair comparisons for new T2I models in the future.
2.	GPT-4o may not be good at handling spatial relationships or counting. The authors only calculate the overall consistency of GPT-4o with human annotators, but do not analyze the performance of GPT-4o on specific categories. There should be more analysis on how GPT-4o performs on different types of concepts.
3.	The validation of GPT-4o as an effective evaluation metric is not sufficient. The authors only compare it with T2VScore in the appendix. It should be compared with other metrics in previous T2I benchmarks and compositional T2I benchmarks.
4.	GPT-4o may be costly as an evaluation metric.

**Strengths:**

1.	The submission presents a novel benchmark, ConceptMix, which addresses limitations in existing compositional T2I evaluations by introducing a more diverse and complex approach to generating and evaluating text prompts in compositionality.
2.	The paper provides detailed definition and combination of different visual concepts for enlarging compositionality benchmark, which is a good supplementary for existing compositional T2I benchmarks. It offers insight for a scalable and customizable solution for evaluating the compositionality.
3.	The benchmark employs discriminative metrics that identify differences in T2I generation models' compositionality. This allows for a more nuanced understanding of model performance and helps specific areas for improvement.

**Additional Feedback:**

N.A.

**Documentation:**

Line165-166 “we generate 300 text prompts to capture the variability and performance across different models”. It seems that there is only template for generation, but no information about the 300 text prompts, which is important for reproduction in a benchmark. A link or a .txt file containing the 300 text prompts should be provided.

**Ethics:**

There are no ethical concerns.

**Limitations:**

The authors addressed the limitations and potential negative societal impact of their work.

**Opportunities For Improvement:**

1.	As a new evaluation metric,validating its effectiveness and consistency with human annotators is critical. In the current version, the effectiveness of the proposed evaluation metric based on GPT-4o is not validated. Currently the consistency with human annotators and a comparison with T2Vscore is in the appendix. I suggest making more comprehensive analysis of how GPT-4o performs for different concepts and comparison between GPT-4o with previous metrics and move those to the main paper. The contributions of the paper can be strengthened with more comprenhensive analysis and stronger validations of the evaluation metrics.

2.	GPT-4o may be costly as an evaluation metric. The authors should consider discussing any cheaper alternative methods for evaluation that could replace GPT-4o, such as using other LLMs that might offer similar capabilities at a lower cost. Also it is expected to discuss the cost of evaluating a model with GPT-4o.

3.	The authors should discuss how to apply GPT-4o as a consistent metric if in the future GPT-4o keeps evolving and we cannot access previous versions of GPT-4o.

**Relation To Prior Work:**

The paper discusses the differences between the work and previous compositional T2I benchmarks in Table 1 and related works.

**Summary And Contributions:**

The paper proposes ConceptMix, a benchmark to evaluate compositional Text-to-Image (T2I) models. Unlike existing evaluations that rely on fixed templates, ConceptMix uses GPT-4o to generate diverse text prompts and for automated grading of visual concepts in generated images. The benchmark demonstrates higher discrimination power, revealing significant performance drops in models with increased complexity (k different visual concepts). CONCEPTMIX is extendable to more visual concepts and offers insights into prompt diversity.

---

> ### Author Rebuttal · Authors · 2024-08-16
>
> We sincerely appreciate your constructive and very thorough feedback. We address the concerns raised below.
>
> 1. **Cons 1 (+ improvement 3):** Consistency & version control of GPT-4o for generating and grading prompts.
>
> While it’s true that GPT will continue to evolve, we believe that this evolution can be an advantage rather than a limitation. As GPT improves its vision understanding capabilities, the correctness evaluation of the generated images will become more accurate, aligning closer to the real-world interpretations. This would make future comparisons even more meaningful.
> Additionally, our primary focus is on the compositional capability of T2I models, more specifically, binding k visual concepts with an object. The consistent evaluation of the compositional capability does not necessarily rely on a static version of GPT but rather on the ability of evaluating increasingly complex and accurate compositions. To verify this, we run additional evaluation experiments using a different VLM (deepseek-vl-7b-chat), we kindly refer the reviewers to the results provided in the **general response (2.)**.
>
> 2. **Cons 2 (+improvement 1):** Human eval with specific concept categories.
>
> To address this, we further conducted a more detailed analysis of GPT-4o's performance across various concept categories by calculating the average consistency scores between the results from GPT-4o and the majority votes from human annotators. Following the human evaluation study settings provided in the paper, we experimented with DALLE 3 (k=1 to 7) and k=3 for all models. Below are the average consistency score of human majority vote and GPT-4o grading results across different concept categories:
>
>
>
> | Category | Consistency (%) |
> |----------|-------------------------|
> | object | 90.86 |
> | color | 86.21 |
> | number | 82.78 |
> | shape | 79.61 |
> | size | 76.92 |
> | texture | 76.03 |
> | style | 74.22 |
> | spatial | 73.33 |
>
>
> These results show that GPT-4o performs relatively well across different categories, with the highest consistency observed in the "object" (90.86%) and "color" (86.21%) categories. However, as expected, the consistency is lower in categories such as "spatial" and "style," which involve more complex spatial reasoning and style recognition tasks which is also challenging to human participants. Other categories like shape (79.61%), size (76.92%), texture (76.03%) fall between these extremes.
>
> 3. **Cons 3 (+improvement 1):** Validate of GPT-4o as an evaluation metrics & compare with previous T2I approaches.
>
> We would like to clarify that aside from T2VScore, we conducted a comprehensive comparison with T2I-compbench, which is widely recognized as one of the most prominent compositional-T2I benchmarks, as shown in Fig. 7 of our paper.  We believe that the comparisons with both T2VScore and T2I-compbench provide a substantial basis for evaluating GPT-4o's effectiveness against established metrics.
>
>
> Moreover, we placed significant emphasis on human evaluation studies to evaluate the effectiveness of our benchmark. While comparing with other T2I benchmarks can offer valuable insights, it is not the sole or definitive method to evaluate the effectiveness of automatic metrics, and we believe that human judgment remains the gold standard despite its limitations of inconsistency among evaluators. We further show that our grading approach is more consistent with human evaluation compared with previous approaches in Fig. 10 & 11. Following the reviewer's suggestions, we will include the human studies in the main paper.
>
>
> 4. **Cons 4 (+improvement 2):** GPT-4o Cost.
>
> We appreciate your concern regarding the cost of using GPT-4o as an evaluation tool. It is true that GPT-4o is not without cost, but we would like to highlight a few important considerations.
>
> Firstly, while open-source model alternatives exist, they currently fall short of GPT-4o's performance, particularly when evaluating complex and compositional image generation tasks. Using less effective models could compromise the quality of the evaluation.
>
> Since we only need to generate 300x7 images per model in our current settings, and considering the fact that new image generation models are not released frequently, the overall cost remains feasible within our research budget.  Detailed cost breakdown: [OpenAI API Pricing](https://openai.com/api/pricing/).
>
> **Input cost:**
> - **Image**: for GPT-4o-2024-05-13 version it cost \$0.003825 for the 1024x1024 res images, the largest size in our experiment, and ther models generate smaller images (512x512 or less).
> - 300(#samples) x 7(#k) x 8(#models) x \$0.003825 = \$64.26
>
> - **Text**: 20 words maximum per question. 8 questions per image (maximum), 1 token ≈ ¾ words, so 20 words ≈ 27 tokens.
> - 27 tokens x 8 questions = 216 tokens per image. 16,800 images x 216 tokens = 3,628,800 tokens.
> - 3,628,800 tokens x \$2.50/1M tokens = \$9.07
>
> **Output cost:**
> - Assuming each yes/no answer is about 1 token. 8 questions (maximum) x 16,800 images = 134,400 tokens.
> - 134,400 tokens x \$7.50/1M tokens = \$1.01
>
> **Total:**
> $74.34 for our entire evaluation experiments (8 models).
>
> It is also worth comparing the cost of GPT-4o to that of human evaluation studies. Human studies are significantly more expensive and time-consuming. For context, our human study cost $660 ($15 per person per hour) and required considerable time to organize and conduct. In contrast, using GPT-4o to evaluate a substantial set of images is considerably more cost-effective and can be completed much faster. Moreover, the cost of using the GPT-4o API has been decreasing over time (e.g., 5.00 dollars per 1M input tokens for GPT-4o-2024-05-13, but 2.50 dollars per 1M input tokens for GPT-4o-2024-08-16), making it an increasingly affordable option.

---

> > ### Comment · Reviewer_UaRX · 2024-08-24
> > **Official Comment by Reviewer UaRX**
> >
> > Thank the authors for the rebuttal. The authors have addressed my questions except for the version control and consistency of GPT-4o as an evaluation metric. I agree that better future versions of GPT-4o will enhance the prompt generation and evaluation in this work. However, the evaluation results in this work may also mislead follow-up works. If researchers compute the GPT-4o scores with future versions of GPT-4o and compare the results with the numbers reported in this paper, this will lead to unfair comparison and benchmarking. This means that future researchers will need to re-evaluate all T2I models on future GPT-4o versions for a fair comparison instead of running their model and directly comparing the results with the results reported in this paper. I suggest the authors emphasize the GPT-4o version and date of evaluation in the paper and prevent future researchers from misunderstanding and misusing the reported results to make unfair comparisons in the future. I increased my rating to borderline reject.

---

> > > ### Author Rebuttal · Authors · 2024-08-28
> > >
> > > We really appreciate your thoughtful feedback and we also want to thank you for raising the score! We will prominently emphasize the specific GPT-4o version (GPT-4o-2024-05-13) and evaluation date (May, 2024) used for our study in the paper. We will add a clear disclaimer in the paper cautioning future researchers about directly comparing their results with ours if using newer versions of GPT-4o. We commit to maintaining the leaderboard and re-evaluating all models using the same, current version of GPT for fair comparisons. Regarding the feasibility of re-evaluation, we'd like to note that the evaluation cost is relatively modest. For example, the current grading cost is $9.29 per model for all k (300 samples each). With future GPT versions, we expect this cost may decrease further, making regular re-evaluation even more practical. Please let us know if you have any other questions or concerns, thank you!

---

### Author Rebuttal · Authors · 2024-08-16

Dear reviewers,

We would like to express our sincere gratitude to all the reviewers for their insightful and thorough feedback! We are delighted that reviewers found our paper *well-written and easy to follow* (Reviewer a4xv), and recognized the *novelty* of our benchmark (Reviewer UaRX) as well as *our careful consideration and diverse and complex approach for concept selection and prompt generation* (Reviewer UaRX, a4xv). We are motivated by comments that our analyses are *interesting, thorough, and insightful* (Reviewer a4xv, r238, UaRX). We will incorporate all the valuable feedback into the final version of our paper. We begin by addressing the common concerns raised by the reviewers:

1. **Writing**: To improve readability, we have integrated the majority of the footnote content directly into the main paper, keeping only 2 footnotes (previously numbered 3 and 6). Additionally, we have moved the human studies from the appendix to the main body of the paper and highlighted the findings in the introduction. We also moved the github link to the main paper.

2. **Additional VLM grading experiments:** We run additional evaluation experiments using different VLMs beyond GPT-4o. We show experimental results with a different VLM - deepseek-vl-7b-chat[1], and we observe that the relative results and the general trend (performance comparison across different models and across different k) still hold true ignoring the specific model.

Here we show the grading results with deepseek-vl-7b-chat:

| Model | k = 1 | k = 2 | k = 3 | k = 4 | k = 5 | k = 6 | k = 7 |
|-------|-------|-------|-------|-------|-------|-------|-------|
| SD v1.4 | 0.61 | 0.32 | 0.15 | 0.09 | 0.03 | 0.02 | 0.00 |
| SD v2.1 | 0.64 | 0.38 | 0.23 | 0.14 | 0.07 | 0.03 | 0.02 |
| DeepFloyd IF XL v1 | 0.73 | 0.48 | 0.31 | 0.17 | 0.10 | 0.05 | 0.01 |
| SDXL Turbo | 0.74 | 0.53 | 0.32 | 0.17 | 0.09 | 0.05 | 0.03 |
| SDXL Base | 0.74 | 0.54 | 0.27 | 0.16 | 0.12 | 0.05 | 0.03 |
| PixArt alpha | 0.76 | 0.48 | 0.38 | 0.20 | 0.11 | 0.08 | 0.04 |
| Playground v2.5 | 0.81 | 0.59 | 0.41 | 0.20 | 0.14 | 0.08 | 0.06 |
| DALL·E 3 | 0.90 | 0.74 | 0.62 | 0.41 | 0.33 | 0.28 | 0.18 |

Both evaluations (GPT-4o & Deepseek-vl) consistently rank DALL·E 3 as the top performer across all k values, with SD v1.4  and SD v2.1 performing the worst. DALL·E 3 maintains a significant lead, particularly at k=3 (0.62 with deepseek-vl, 0.50 with GPT-4o). The relative ranking of models remains stable. All models show a clear performance decline as k increases. For instance, DALL-E 3 scores 0.90 at k = 1 and 0.18 at k = 7 with deepseek-vl-7b-chat, while in our GPT-4o results, it scores 0.83 at k = 1 and 0.08 at k = 7. Similarly, Playground v2.5 scores 0.81 at k = 1 and 0.06 at k = 7 with deepseek-vl-7b-chat, compared to 0.70 at k = 1 and 0.01 at k = 7 with GPT-4o. Notably, Deepseek-vl evaluations show slightly higher numbers than GPT-4o evaluations across the board. We have included figures in the **pdf attached below (Fig. 2)** to provide a clearer visualization of the general trends.

[1] Lu, Haoyu, et al. "Deepseek-vl: towards real-world vision-language understanding." arXiv 2024.


3. **The reliability of Concept Evaluation step:** Concept evaluation questions were created automatically (lns. 134-149, appendix lns. 627-630). To confirm their effectiveness, we analyzed 100 randomly sampled prompts to identify cases where concepts mentioned in the prompts were not given in the selected visual concepts. Only one case (1%) showed a mismatch between the selected concepts and the generated concept evaluation questions. For instance:

Visual concepts: pine tree, bee, tiny, photorealism, tiny, metallic, top, left.

Prompt: A tiny pine tree on the left side of the image has a tiny metallic bee positioned on top of it. The scene is depicted in a photorealistic style.

The questions are:

- Does the image contain a pine tree?
- Does the image contain a bee?
- Is the pine tree tiny in size?
- Is the style of the image photorealism?
- Is the bee tiny in size?
- Does the bee have a metallic texture?
- Is the bee on top of the pine tree?
- Is the pine tree positioned on the left side of the bee?

The final question, Is the pine tree positioned on the left side of the bee? inaccurately interprets "left side of the image" as relative to the bee. This single instance of misalignment suggests that the overall concept representations in the prompts and questions are highly accurate, with only minor, isolated discrepancies. In this context, such rare occurrences are negligible and unlikely to significantly impact the evaluation.


Below we address the remaining comments from the reviewers.

---

### Decision · Program_Chairs · 2024-09-26

**Decision:**

Accept (Poster)

**Comment:**

Congratulations!

Multiple reviewers appreciated the depth of the ConceptMix benchmark, especially for its ability to handle complex compositional prompts and provide detailed evaluation of text-to-image (T2I) models. The benchmark’s discrimination power and scalability were also appreciated, along with its analyses of 8 models.

However, there are still concerns about the reliance on GPT-4o for evaluation, particularly due to transiency, which could lead to future inconsistencies in benchmarking. The reviewers suggested validating the method with alternative models and adding more human annotations to strengthen the evaluation. In the end, these concerns were not completely addresses. I could have seen a much stronger version of this paper—one that could get an oral—if these concerns were addressed. With the reliance on GPT-4o, I don't really see how the results from one run could be comparable with another researcher's evaluation in a few months.

Despite these critiques, the paper was generally well-received.